# Organic Compounds as Corrosion Inhibitors for Carbon Steel in HCl Solution: A Comprehensive Review

**DOI:** 10.3390/ma15062023

**Published:** 2022-03-09

**Authors:** Liangyuan Chen, Dongzhu Lu, Yanhu Zhang

**Affiliations:** 1Open Studio for Marine Corrosion and Protection, Pilot National Laboratory for Marine Science and Technology, No. 1 Wenhai Road (Qingdao), Qingdao 266200, China; chen_ly2020@163.com; 2Institute of Oceanology, Chinese Academy of Sciences, No. 7 Nanhai Road, Qingdao 266071, China; 3Institute of Advanced Manufacturing and Modern Equipment Technology, Jiangsu University, Zhenjiang 212013, China

**Keywords:** evaluation method, corrosion inhibition performance, 1.0 M HCl, carbon steel, statistical analysis

## Abstract

Most studies on the corrosion inhibition performance of organic molecules and (nano)materials were conducted within “carbon steel/1.0 M HCl” solution system using similar experimental and theoretical methods. As such, the numerous research findings in this system are sufficient to conduct comparative studies to select the best-suited inhibitor type that generally refers to a type of inhibitor with low concentration/high inhibition efficiency, nontoxic properties, and a simple and cost-economic synthesis process. Before data collection, to help readers have a clear understanding of some crucial elements for the evaluation of corrosion inhibition performance, we introduced the mainstay of corrosion inhibitors studies involved, including the corrosion and inhibition mechanism of carbon steel/HCl solution systems, evaluation methods of corrosion inhibition efficiency, adsorption isotherm models, adsorption thermodynamic parameters QC calculations, MD/MC simulations, and the main characterization techniques used. In the classification and statistical analysis section, organic compounds or (nano)materials as corrosion inhibitors were classified into six types according to their molecular structural characteristics, molecular size, and compound source, including drug molecules, ionic liquids, surfactants, plant extracts, polymers, and polymeric nanoparticles. We outlined the important conclusions obtained from recent literature and listed the evaluation methods, characterization techniques, and contrastable experimental data of these types of inhibitors when used for carbon steel corrosion in 1.0 M HCl solution. Finally, statistical analysis was only performed based on these data from carbon steel/1.0 M HCl solution system, from which some conclusions can contribute to reducing the workload of the acquisition of useful information and provide some reference directions for the development of new corrosion inhibitors.


**Contents**

1. Introduction22. Corrosion and inhibition mechanism of carbon steel/HCl solution systems4 2.1. Corrosion mechanism4 2.2. Inhibition mechanism53. Experimental and theoretical research methods and characterization techniques7 3.1. Inhibition performance evaluation methods and experimental parameters7  3.1.1. Weight loss measurements (WL)8  3.1.2. Potentiodnamic polarization (PDP)8  3.1.3. Electrochemical impedance spectroscopy (EIS)9  3.1.4. Electrochemical frequency modulation (EFM)10  3.1.5. Other evaluation methods9 3.2. Adsorption isotherms and Thermodynamic parameters12  3.2.1. Adsorption isotherms12  3.2.2. Thermodynamic parameters13 3.3. Theoretical calculations14  3.3.1. DFT calculations15   (1) Global reactivity descriptors15   (2) Local reactivity descriptors17  3.3.2. MD and MC simulations20 3.4. Physicochemical and morphology characterization techniques244. Classification and statistical analysis25 4.1. Drug molecules25 4.2. Ionic liquids28 4.3. Surfactants30 4.4. Plant extracts33 4.5. Polymers and Polymeric-nanoparticles37 4.6. Statistical analyses425. Conclusions45References47

## 1. Introduction

Corrosion is a common problem for industrial metals and directly impacts their cost and safety. Presently, there are many ways to retard metal corrosion, such as the optimization of the metal constituents and smelting process, organic/inorganic coating technology, and the addition of corrosion inhibitors, among which the addition of corrosion inhibitors is the most economical and commonly used [1,2,3,4,5,6,7,8,9,10]. From many literature surveys, the research on corrosion inhibition behavior for carbon steel has been found to be most common, firstly because of carbon steel with its relatively high strength, low cost, and widespread availability in numerous industrial fields, and second because the corrosion of carbon steel is a common phenomenon during their production and applications due to basic properties of iron (a highly reactive material), particularly in chemical and petrochemical industries, carbon steel are in direct contact with strong acidic solution, resulting in the reduction of their service life and even causing serious accidents [11,12]. The media used to study the corrosion inhibition ability of corrosion inhibitor in different literature were mainly 1.0 M HCl solution, which is primarily because most metal chlorides are easily soluble in water. The treatment of the rusty metal with HCl solution is capable of dissolving a wide range of corrosion products on metal, while the HCl solution is used for chemical cleaning processes as an acid detergent in which the concentration is exactly 1.0 M. Moreover, in carbon steel/HCl solution systems, the reaction rate of iron oxide with HCl is over three times faster than that of iron oxide with H_2_SO_4_, which is far more than that of iron oxide with HNO_3_, HClO_4_, citric acid, formic acid, and acetic acid [13]. Therefore, the use of efficient, safe, and low-cost HCl as an acid detergent is well recognized. On the other hand, it is important to note that the corrosion rate of carbon steel in HCl is more than twice that of carbon steel in H_2_SO_4_ and choosing the right corrosion inhibitor is crucial during the pickling process [14].

Corrosion inhibitor selection is based on the metal substrate and the surrounding environment (solvent nature, temperature, and pH of solution, etc.), while at the same time, their economy, efficacy, and environmental factors should be considered. Currently, finding inexpensive, easy to synthesize, highly efficient, and nontoxic inhibitors is a challenge. According to the literature, corrosion inhibitors fall into two general categories, i.e., organic compounds and inorganic counterparts, while organic compounds as corrosion inhibitors may in part meet the above requirements compared to some inorganic counterparts as inhibitors, such as phosphate and nitrate [15,16]. Additionally, their inhibition mechanisms are distinct; organic compounds can physically or chemically or both interact with metal surfaces and limit the cathodic, anodic, or both reaction rates by blocking the active sites. This is already explained in the literature, in which physical interaction refers to inhibitor molecule adsorption on a metal surface through van der Waals forces or Coulombic forces between them, while chemical interaction means that sulfur (S), nitrogen (N), phosphorus (P), oxygen (O), and conjugated groups (functional groups, heteroatoms, and benzenoid and nonbenzenoid multiple bonds, etc.) contained within organic inhibitors act as adsorption centers that link the inhibitor (donating electron) to the metal surface (accepting electron) [17]. Both the surface adsorption of molecules and the bonding between them play a role in inhibiting corrosion. For inorganic salts, the corrosion process is inhibited mainly by passivating the metal’s surface to form a protective oxide film against the corrosive environment [18,19].

Organic compounds as corrosion inhibitors were developed in the petroleum industry in the 1950s and subsequently used widely [20]. Currently, many types of organic compounds as corrosion inhibitors have been considered to have potential applicability, including drug molecules, ionic liquids, surfactants, plant extracts, polymers, the combination and modified structure of two or more compounds, and the combination of these compounds and inorganic salts or polymeric nanoparticles. It has been recognized in multiple published studies that the use of these organic inhibitors is an effective method for protecting carbon steel from corrosion, while a significant proportion of these studies explored organic inhibitors’ inhibition efficiency and inhibition mechanism within carbon steel/1.0 M HCl solution system. However, there is still a need for a review study to obtain a contrast between numerous research results under this system to overcome the limitations of individual studies and reduce the workload of the acquisition of useful information.

Alternatively, there are many different experimental techniques and theoretical analysis methods that have been used to enhance the validity of the findings about inhibition efficiency, inhibition mechanism, and the molecular structure of an inhibitor. For example, the experimental methods used for investigating the inhibition efficiency include weight loss (WL), potentiodynamic polarization (PDP), electrochemical impedance spectroscopy (EIS), hydrogen evaluation (HE), and so on. The theoretical analysis methods used for investigating the inhibition mechanism include adsorption isotherms, thermodynamic parameters, quantum mechanical (QM) calculations, molecular dynamics (MD), and Monte Carlo (MC) simulations, and the corresponding model and model parameters. The characterization techniques used for investigating physico-chemical structures include Fourier transform infrared spectroscopy (FTIR), UV–visible spectroscopy (UV), X-ray diffraction spectroscopy (XRD), mass spectrometry (MS), energy dispersive spectrometry (EDS), atomic force microscopy (AFM), and so on.

This review article will first briefly describe the different experimental techniques and theoretical analysis methods widely used for corrosion inhibitor research in the last few years. Then, organic compounds or (nano)materials as corrosion inhibitors will be classified according to their molecular structural characteristics, molecular size, and compound source, and important conclusions obtained from recent literature will be outlined. Finally, statistical analysis will be only carried out using data from carbon steel/1.0 M HCl solution system. Specifically, the critical information and contrastable experimental data of each type of corrosion inhibitor from different studies will be enumerated and listed and we will compare the corrosion inhibition property differences among these types of inhibitors, especially in their maximum inhibition efficiency, optimum concentration, and optimum temperature. This work may be helpful to provide a clearer choice of anti-corrosion options for this system in practice work. At the same time, it can also provide some reference directions for the development of new corrosion inhibitors.

## 2. Corrosion and Inhibition Mechanism of Carbon Steel/HCl Solution Systems

### 2.1. Corrosion Mechanism

Metal corrosion is a localized electrochemical reduction–oxidation reaction occurring on its surface, in which electrons are released due to metal dissolution and transferred to a different location on the surface to reduce hydrogen ions. This process results in the slow degradation and eventual failure of the metal. Prior to the discussion of various corrosion inhibition evaluation methods, it is necessary to understand the underlying principles of corrosion in carbon steel/HCl solution systems. Like many other metals, the iron corrosion process can also be broken down into two main half electrochemical reactions [21,22,23], where one is the anodic reaction (oxidative dissolution of iron). The overall chemical reaction of iron immersed in HCl solutions is summarized as shown in Equation (1), while the anodic reactions of iron immersed in aqueous solutions and aqueous solutions containing Cl^−^ ions are summarized as shown in Equations (2)–(7) [24,25].

(i) HCl solutions (overall chemical reaction):(1)$$\mathrm{Fe}(\mathrm{solid})+{2H}^{+}\to {\mathrm{Fe}}^{2+}+H_{2}\left(\mathrm{gas}\right)$$

(ii) Aqueous solutions (oxidative dissolution):(2)$$\mathrm{Fe}+H_{2}O\Leftrightarrow {\left[\mathrm{FeOH}\right]}_{\mathrm{ads}}^{}+H^{+}+e^{-}$$
(3)$${\left[\mathrm{FeOH}\right]}_{\mathrm{ads}}\Leftrightarrow {\left[\mathrm{FeOH}\right]}_{\mathrm{ads}}^{+}+e^{-}\;(\mathrm{rate}\;\mathrm{determining}\;\mathrm{step})\;$$
(4)$${\left[\mathrm{FeOH}\right]}_{\mathrm{ads}}^{+}+H^{+}\Leftrightarrow {\mathrm{Fe}}^{2+}+H_{2}O$$

(iii) Aqueous solutions containing Cl^−^ ions (oxidative dissolution):(5)$$\mathrm{Fe}+H_{2}O+{\mathrm{Cl}}^{-}\Leftrightarrow {\left[\mathrm{FeClOH}\right]}_{\mathrm{ads}}^{-}+H^{+}+e^{-}$$
(6)$${\left[\mathrm{FeClOH}\right]}_{\mathrm{ads}}^{-}\Leftrightarrow {\left[\mathrm{FeClOH}\right]}_{\mathrm{ads}}+e^{-}\;(\mathrm{rate}\;\mathrm{determining}\;\mathrm{step})\;$$
(7)$$\left[\mathrm{FeClOH}\right]+H^{+}\Leftrightarrow {\mathrm{Fe}}^{2+}+{\mathrm{Cl}}^{-}+H_{2}O$$

It can be seen that iron exposed to the above solutions tends to dissolve and lose positive Fe ions to the electrolyte, which simultaneously produces free electrons that can travel through the metal. [FeOH]_ads_ and [FeClOH]_ads_ are the adsorbed intermediates, each of which is involved in the rate determining step of Fe dissolution according to mechanisms (ii) and (iii). It must be pointed out that the presence of Cl^−^ ions does not exclude dissolution through the [FeOH]_ads_ intermediate in chloride free acid media, as the two mechanisms can proceed simultaneously [26]. Gad Allah et al. [27] pointed out that iron dissolution in HCl solutions depends on H^+^ ions more than Cl^−^ ions. According to Oakes and West [28], iron dissolution in HCl solutions over the pH range 0.0 to −0.6 (as 1.0 M HCl solution) depends principally upon chloride ion activity, while at more negative pH values and at high chloride ion activity, the corrosion rate is more dependent upon pH.

On the other hand, for an acidic solution, the electric potential is caused by the accumulation of excess electrons generated in the anode, which can be neutralized at the cathodic site by the reduction of H^+^ to form hydrogen gas. This process can be presented as follows (Equations (8)–(10)) [25].
(8)$$\mathrm{Fe}+H^{+}\Leftrightarrow {\left(\mathrm{FeH}\right)}_{\mathrm{ads}}^{+}$$
(9)$${\left(\mathrm{FeH}\right)}_{\mathrm{ads}}^{+}+e^{-}\Leftrightarrow {\left(\mathrm{FeH}\right)}_{\mathrm{ads}}$$
(10)$${\left(\mathrm{FeH}\right)}_{\mathrm{ads}}+H^{+}+e^{-}\Leftrightarrow \mathrm{Fe}+H_{2}$$

Ehteram and Aisha obtained corrosion rates for mild steel samples at different concentrations of HCl solutions at 25 °C through hydrogen evolution and mass loss measurements, and their results are shown in Table 1 [24]. As expected, both *ρ*_HE_ and *ρ*_ML_ increase with increasing HCl concentration, indicating acceleration behavior for mild steel dissolution. Meanwhile, they demonstrated that the relationship between corrosion rates and HCl concentration fits the following relation (Equation (11)) [29].
(11)$$\mathrm{lg}\rho =\mathrm{lg}k+B\mathrm{lg}c_{HCl}$$
where *k* is the specific reaction rate constant, *B* is the reaction constant, and *c*_HCl_ is the amount of concentration of HCl solution. The *k* value represents the corrosion rate when the acid concentration is equal to unity.

### 2.2. Inhibition Mechanism

In HCl solutions, added organic compounds can form a thin layer on the metal surface and significantly reduce the corrosion rate, which is regarded as a substitution reaction that occurs between inhibitor molecules and water molecules at the metal/solution interface, which can be described as below (Equation (12)) [30,31,32]:(12)$${\mathrm{Org}}_{\left(\mathrm{sol}\right)}+xH_{2}O_{\left(\mathrm{ads}\right)}\Leftrightarrow {\mathrm{Org}}_{\left(\mathrm{ads}\right)}+xH_{2}O_{\left(\mathrm{sol}\right)}$$
where Org_(sol)_ and Org_(ads)_ are inhibitor molecules dissolved in solution and inhibitor molecules adsorbed on the metal surface, respectively, and H_2_O_(sol)_ and H_2_O_(ads)_ are water molecules and adsorbed water molecules on the metal surface, respectively. *x*, the size ratio, represents the number of water molecules displaced by one molecule of organic inhibitor [33,34]. It is noteworthy that the size ratio depends on the geometry of the organic inhibitor. In general, an organic inhibitor with planar geometry provides higher surface coverage and thereby behaves better as a corrosion inhibitor [35,36,37,38].

In the presence of corrosion inhibitors, the adsorbed intermediates accounting for the mitigation of Fe anodic dissolution can be presented as follows (Equations (13)–(19)) [25,32,39,40]:(13)$$\mathrm{Fe}+H_{2}O\Leftrightarrow {\mathrm{FeH}}_{2}O_{\mathrm{ads}}$$
(14)$${\mathrm{FeH}}_{2}O_{\mathrm{ads}}+\mathrm{INH}\Leftrightarrow {\mathrm{FeOH}}_{\mathrm{ads}}^{-}+H^{+}+\mathrm{INH}$$
(15)$${\mathrm{FeH}}_{2}O_{\mathrm{ads}}+\mathrm{INH}\to \mathrm{Fe}-{\mathrm{INH}}_{\mathrm{ads}}+H_{2}O$$
(16)$${\mathrm{FeOH}}_{\mathrm{ads}}^{-}\to {\mathrm{FeOH}}_{\mathrm{ads}}+e^{-}\;(\mathrm{rate}\;\mathrm{determining}\;\mathrm{step})\;$$
(17)$$\mathrm{Fe}-{\mathrm{INH}}_{\mathrm{ads}}\to \mathrm{Fe}-{\mathrm{INH}}_{\mathrm{ads}}^{+}+e^{-}$$
(18)$${\mathrm{FeOH}}_{ads}+\mathrm{Fe}-{\mathrm{INH}}_{\mathrm{ads}}^{+}\Leftrightarrow \mathrm{Fe}-{\mathrm{INH}}_{\mathrm{ads}}+{\mathrm{FeOH}}^{+}$$
(19)$${\mathrm{FeOH}}^{+}+H^{+}\Leftrightarrow {\mathrm{Fe}}^{2+}+H_{2}O$$

According to the above reaction process, the dissolution of Fe in acid solution depends mainly on the adsorbed intermediate species, wherein the reduction in the amount of ${\mathrm{FeOH}}_{\mathrm{ads}}^{-}$ produced (Equation (14)) due to replacement of H_2_O with inhibitor molecules (Equation (15)) (the formation of intermediate Fe–INH_ads_) retards the rate determining step (Equation (16)) and consequently retard the dissolution of Fe. However, it needs to be emphasized that, in most cases (in acidic media), adsorption of these inhibitor molecules tends to be initiated by using physisorption and then propagated by using a chemisorption mechanism, i.e., a physiochemisorption mechanism [41,42]. Their mechanism is briefly described as follows.

Physisorption mechanism: excessive oxidation of Fe elements in the HCl solution makes the carbon steel surface positively charged, which attracts negatively charged chloride ions, causing the surface to be negatively charged and forming a so-called inner Helmholtz plane (IHP) (as shown in Equation (20)), while the attractive forces between the positively charged inhibitor molecules INH–H^+^ (because heteroatoms become protonated in aggressive acidic media) and the carbon steel surface increased as a result of a bridge created by the adsorption of chlorides (Cl^−^), which formed the outer Helmholtz plane (OHP) [32]. It has been reported that such physical interactions between the inhibitor and the iron surface would behave loosely with increasing temperature [43].
(20)$${\mathrm{Fe}(H}_{2}{O)n}_{\mathrm{ads}}+{2\mathrm{Cl}}^{-}\Leftrightarrow \mathrm{Fe}{\left[{{(H}_{2}O)}_{n}{\left(\mathrm{Cl}\right)}_{2}^{-}\right]}_{\mathrm{ads}}^{}$$

Chemisorption mechanism: further oxidation of surface iron atoms results in the production of electrons that are consumed by INH−H^+^, so the adsorbed cationic inhibitor molecules return their neutral form (as shown in Equation (21)), while heteroatoms with lone pairs of electrons can transfer their lone pair of electrons into the d-orbitals of the surface iron atoms, which results in chemisorption. Such chemical interactions between the inhibitor and the surface are stronger than the physical interactions between them [32].
(21)$$\mathrm{Fe}{\left[{{(H}_{2}O)}_{n}{(\mathrm{Cl}}_{2}^{-})\right]}_{\mathrm{ads}}+\mathrm{INH}-H^{+}\to \mathrm{Fe}{\left[{{(H}_{2}O)}_{n}{\left(\mathrm{Cl}\right)}_{2}\mathrm{INH}-H\right]}_{\mathrm{ads}}^{-}$$

The abovementioned electron transfer causes electron accumulation in the d-orbitals of iron atoms, which in turn can cause a reverse transfer of electrons from the d-orbitals of surface iron atoms to the unoccupied anti-bonding molecular orbitals of inhibitor molecules due to interelectronic repulsion, i.e., retro-donation mechanism. The greater donation of electrons can lead to greater retrodonation, and both of them can strengthen each other through synergism [44,45,46]. In 1.0 M HCl medium, a diagrammatic illustration of the three adsorptions of organic corrosion inhibitors (DHATs) is presented in Figure 1 [45].

The substituent in organic corrosion inhibitor molecules can not only affect coverage but also the electron density over the active sites. Generally, organic molecule with electron-releasing substituents such as –NH_2_, –OH, –OCH_3_, –CH_3_, etc. act as better corrosion inhibitors than that with electron withdrawing substituents such as –NO_2_, –CN, –COOH, etc. [32]. In the case of heterocyclic compounds, the adsorption process onto metal surfaces can be achieved through their electron-rich centers (polar functional groups: –NHMe, –NH_2_, –NMe_2_, –OH, –NO_2_, –OCH_3_, –O–, –CN, –CONH_2_, –COOC_2_H_5_, etc.), and/or π-electrons of the heteroatomic (>C=O, >C=N–, >C=S, –C≡N, –N=O, and –N=S, etc.), and/or homoatomic (>C=C<, –N=N–, and –C≡C–, etc.) [47,48,49]. In addition to this electronic effect, the introduction of nonpolar hydrophobic chains such as –CH_3_ can hinder the adsorption of polar electrolyte molecules over the metallic surface due to their hydrophobic action (low solubility), while polar substituents such as –OH, –NH_2_, –CN and –NO_2_, etc. can enhance the solubility of inhibitor in HCl solutions and other polar electrolytes, whereas they can play an opposing role in the adsorption of polar electrolyte molecules [50,51]. Therefore, the electronic effect and hydrophobic action may cancel each other out or reinforce each other. In terms of corrosion inhibition, it is hard to determine which dominated between the two effects. 

## 3. Experimental and Theoretical Research Methods and Characterization Techniques 

### 3.1. Inhibition Performance Evaluation Methods and Experimental Parameters

For a more accurate comparison of the corrosion inhibition efficiency of inhibitors under different environments (or different classes or concentrations), multiple distinct experiments are usually used to evaluate inhibition efficiency. Some of the methods used to evaluate corrosion inhibition efficiency are listed in Table 2. Among them, WL, PDP, and EIS are the most fundamental and the most effective evaluation methods for corrosion resistance. Likewise, the three methods are also the most frequently used for evaluating corrosion inhibition efficiency.

#### 3.1.1. WL Measurements

As is well known for most WL measurements studies, the corrosion rate (*C*_WL_) should be first computed by using the following Equation (22). Then, based on the calculation results, Equations (23) and (24) can be further used to calculate the inhibition efficiency (IE) of the studied inhibitor and its surface coverage (*θ*) [52,53].
(22)$$C_{\mathrm{WL}}=\frac{K\times W}{A\times t\times \rho }$$

For carbon steel, *ρ* = 7.86 g·cm^−3^, K = 8.76 × 10^4^, W indicates the mass loss in grams, t is the immersion time of the specimens inside the corrosion medium (in hours or days), and the exposed area is represented as A in cm^2^.
(23)$$\mathrm{EI}(\%)=\frac{C_{\mathrm{WL}}-C_{\mathrm{WL}(\mathrm{inh})}}{C_{\mathrm{WL}}}\times 100$$
(24)$$\theta =\frac{C_{\mathrm{WL}}-C_{\mathrm{WL}(\mathrm{inh})}}{C_{\mathrm{WL}}}$$
where *C*_WL_ and *C*_WL(inh)_ signify the corrosion rates of the tested inhibitor at different concentrations, and in the blank, *θ* refers to the surface coverage degree of the studied inhibitor.

The WL method is accurate but time-consuming and is not available for rapidly evaluating the performance of the studied inhibitor. Additionally, the method cannot be used to evaluate localized corrosion; it can only be used for general corrosion. However, a series of electrochemical methods can largely compensate for these shortcomings.

#### 3.1.2. PDP Technique

PDP technique is mainly used to investigate the kinetics of corrosion reaction, and this technique has been used in the majority of corrosion inhibitor studies. In general, the anodic and cathodic branches of the measured polarization curves shift progressively toward the lower current density with increasing concentrations of corrosion inhibitors, which indicates that the overall corrosion rate is decreased. The corrosion potential (*E*_corr_) and corrosion current density (*i*_corr_) can be obtained from the Tafel extrapolation of the polarization curves, in which *i*_corr_ is frequently used to calculate the inhibition efficiency, as in Equation (25) [53]. In addition to the calculation of the inhibition efficiency, the obtained *E*_corr_ is also used for determining the corrosion inhibitor type; that is, if the value of *E*_corr_ > 85 mV compared to the blank solution inhibitor can behave as an anodic or cathodic type, and if the *E*_corr_ value < 85 mV, it will behave as a mixed type of inhibitor [54].
(25)$$\mathrm{EI}(\%)=\frac{i_{\mathrm{corr}}-i_{\mathrm{corr}(\mathrm{inh})}}{i_{\mathrm{corr}}}\times 100$$
where *i*_corr_ and *i*_corr(inh)_ are the measured corrosion current densities in the absence and presence of inhibitors, respectively.

Similarly, the PDAP technique was applied to examine nutmeg oil as a pitting corrosion inhibitor for L-52 carbon steel in 1.0 M HCl by Abdallah et al. [55]. The results showed that *E*_pitt_ moved toward the noble (+) direction as the concentration of nutmeg oil increased, which indicated that the pitting attack was reduced, and nutmeg oil could be used as an excellent pitting corrosion inhibitor for L-52 carbon steel. Moreover, the inhibition efficiency can also be calculated according to the polarization resistance (*R*_p_) obtained by LPR tests, as shown in Equation (26) [56].
(26)$$\mathrm{EI}(\%)=\frac{R_{p(\mathrm{inh})}-R_{p}}{R_{p(\mathrm{inh})}}\times 100$$
where *R*_p_ and *R*_p(inh)_ are the measured polarization resistance in the absence and presence of inhibitors, respectively.

#### 3.1.3. EIS Measurements

EIS is a nondestructive corrosion measurement technique that allows for the analysis of corrosion inhibitor films on metal surfaces [57]. The EIS parameters can be obtained by analyzing the experimental EIS spectra as well as a suitable equivalent circuit, including the solution resistance (*R*_s_), charge transfer resistance (*R*_ct_), electric double layer capacitance (*C*_dl_), film capacitance of the double layer (*C*_c_), Warburg resistance (*W*), and so on. Such parameters may reflect the information about the corrosion resistance, the surface roughness, and the thickness and uniformity of the adsorbed layer [58]. Meanwhile, the total resistance (*R*_t_) of the system is calculated according to these parameters, which can be used to further calculate the inhibition efficiency, as shown in Equation (27) [59].
(27)$$\mathrm{EI}(\%)=\frac{R_{t(\mathrm{inh})}-R_{t}}{R_{t(\mathrm{inh})}}\times 100$$
where *R*_t_ and *R*_t__(in__h)_ are the calculated total resistance of the electrode surface in the absence and presence of inhibitors, respectively. Depending on the equivalent circuit, the value of *R*_t_ can be a simple addition of several resistances, a parallel coupling value of several resistances, or a combination of the two.

#### 3.1.4. EFM Technique

EFM is also a nondestructive corrosion measurement technique that can directly give the values of EFM parameters (including *i*_corr/EFM_ corrosion current density, *β*_a_ *β*_c_ Tafel slopes, CF2, CF3 causality factors) without prior knowledge of Tafel constants, and the inhibition efficiency can be calculated according to Equation (28) [60,61].
(28)$$\mathrm{EI}(\%)=\frac{i_{\mathrm{corr}/\mathrm{EFM}}-i_{\mathrm{corr}/\mathrm{EFM}(\mathrm{inh})}}{i_{\mathrm{corr}/\mathrm{EFM}}}\times 100$$
where *i*_corr/EFM_ and *i*_corr/EFM(inh)_ are the corrosion current densities in the absence and presence of inhibitors, respectively.

#### 3.1.5. Other Evaluation Methods

The corrosion inhibition efficiency of an inhibitor can also be obtained from other different methods. For example, Nnaji et al. [62] estimated the corrosion inhibition efficiency of two corrosion inhibitors over aluminum using FTIR peak intensities (the formula is shown in Equation (29)), where *A*_0_ and *A*_x_ represent the absorbance of the infrared peak at 3300 cm^−1^ (i.e., the signals of Al(OH)_3_, AlOOH, and hydrated aluminum (Al-H_2_O)) for corroded aluminum in the absence and presence of inhibitors [63]. For corrosion protection evaluation of thin film, Kowsari et al. [64] thought that EN technique could be a more reliable method than potentiodinamic polarization and electrochemical impedance spectroscopy. In their study, the inhibition efficiency was calculated based on the time records of electrochemical current noise for mild steel immersed in 1.0 M HCl solution in the absence and presence of tetra-n-butyl ammonium methioninate, as shown in Equation (30), where *R*_n_ is the noise resistance, which is equivalent to the polarization resistance (*R*_p_) measured by linear polarization, and *R*_n(inh)_ and *R*_n_ represent the noise resistances with and without inhibitor [65].
(29)$$\mathrm{EI}(\%)=\frac{A_{0}-A_{x}}{A_{0}}\times 100$$
(30)$$\mathrm{EI}(\%)=\frac{R_{n(\mathrm{inh})}-R_{n}}{R_{n(\mathrm{inh})}}\times 100$$

For the corrosion reaction of carbon steel in HCl solutions, there is a quantitative relationship between corrosion rates and oxygen absorption amount or hydrogen emission amount. Thus, in a sealed container, the corrosion rates can be determined by measuring the changes in gas volume during the corrosion process, and this method is called hydrogen evolution measurements. Obviously, the inhibition efficiency can be obtained by HE, and this method can also be well used to determine whether the cathode process is an oxygen evolution reaction or hydrogen evolution reaction. In a study on three acrylamide ionic liquids as corrosion inhibitors for carbon steel in 1.0 M HCl, Tamany et al. measured hydrogen generation rates (H_r_) in the presence and absence of inhibitors using the water replacement method (as shown in Equation (31), the slope of the relationship between the volume of hydrogen evolved and time for carbon steel in 1.0 M HCl was taken as H_r_), and then their inhibition efficiencies were calculated according to the following formula (Equation (32)). Their results demonstrated that the proposed method could obtain the same effectiveness order of the three inhibitors as PDP and EIS measurements [66].
(31)$$H_{r}=\frac{{(V}_{b}{-V}_{a})}{{(t}_{b}{-t}_{a})}$$
where V_a_ and V_b_ are the hydrogen evolved volumes at times t_a_ and t_b_, respectively.
(32)$$\mathrm{EI}(\%)=\left(\frac{H_{\mathrm{ro}}{-H}_{r}}{H_{\mathrm{ro}}}\right)\times 100$$
where H_ro_ and H_r_ are the hydrogen generation rates in the presence and absence of the inhibitor, respectively.

In addition, the concomitant thermal changes in the corrosion reaction can cause temperature changes in the medium. In some studies, the inhibition efficiency was also calculated by recording the curves of temperature versus time in the presence and absence of the inhibitor. For example, Fouda et al. [67] determined the corrosion inhibition efficiency of an inhibitor (for carbon steel/1.0 M HCl system) based on the percentage reduction in reaction number (RN, as determined by Equation (33)) in the presence of the inhibitor, as shown in Equation (34).
(33)$$\mathrm{RN}=\frac{T_{m}-T_{i}}{t}$$
where *T*_m_ and *T*_i_ are the maximum and initial temperatures, respectively, and t is the time used for reaching *T*_m_.
(34)$$\mathrm{EI}(\%)=\left(\frac{{\mathrm{RN}}_{\mathrm{free}}{-\mathrm{RN}}_{\mathrm{inh}}}{{\mathrm{RN}}_{\mathrm{free}}}\right)\times 100$$
where RN_free_ and RN_inh_ are the reaction numbers in the presence and absence of the inhibitor, respectively.

In addition, several electrochemical techniques can also be employed to investigate the adsorption behavior of inhibitors. For example, CV measurements, according to the change in the current values with respect to the applied potential, are utilized for investigation of the electrochemical oxidation–reduction behavior of an electrochemical interface, which can be used to explore the adsorption and inhibition performance of the inhibitor over the metal surface, but most relevant studies can only be used for obtaining information about the improvement or mitigation of the electrochemical behavior after inhibitor addition, not for obtaining the specific inhibition efficiency value [68,69]. More precisely, CV measurements primarily provide support for the explanation of the inhibition mechanism. Srivastava et al., in a study looking at irbesartan drug molecules as an inhibitor of mild steel corrosion in 1.0 M HCl, according to the comparison of energy difference between the lowest unoccupied molecular orbital (LUMO) energy of inhibitor and the highest occupied molecular orbital (HOMO) energy of iron, as well as LUMO energy of iron and HOMO energy of inhibitor (both LUMO energy and HOMO energy were calculated based on onset reduction potential from CV measurements), found that irbesartan molecules are adsorbed on mild steel surface by electrostatic charges of inhibitor and iron, and then irbesartan molecules formed a chemical bond with iron by electron exchange (i.e., both physical and chemical interactions between inhibitor and mild steel) [70,71,72].

In another study by Qiang et al. [73], ginkgo leaf extract as a corrosion inhibitor for X70 steel in 1.0 M HCl was investigated. By calculating *E*_r_ values as the difference between the potential of zero charge (*E*_PZC_) and open circuit potential (*E*_OCP_) of the X70 steel in the solution (as shown in Equation (35), a negative *E*_r_ value is favorable for the adsorption of cations, while a positive *E*_r_ value indicates the preferential adsorption of anions [74]), and the charge on the steel surface was determined [75]. Their results verify that Cl^−^ ions are first adsorbed on the X70 steel surface, causing a negatively charged surface, and then the main organic constitution of extract adsorbed onto the steel surface through an electrostatic interaction, forming a physical barrier. In a previous study on halogen-substituted imidazoline derivatives as corrosion inhibitors for mild steel in hydrochloric acid solution, based on PZC measurements, similar conclusions were drawn by Zhang et al. [76]. SVET technique has great application potential for corrosion and electrochemistry, and it is also used to evaluate the corrosion inhibition efficiency of inhibitors by several researchers [77]. Qiang et al. [78] evaluated the local corrosion of Q235 steel in 1.0 M HCl using SVET based on the current density mapping in different test solutions and immersion times, with the results showing that anodic current density obviously decreased after the addition of inhibitor (a green antihypertensive drug, losartan potassium) in 1.0 M HCl solution, and the current value remained steady with increasing immersion time, which confirmed the strong absorption of this inhibitor over the Q235 carbon steel surface. In the same metal/electrolyte system, a significant reduction in the anode current density after the addition of a green imidazole inhibitor was also observed using SVET by Yang et al. [79].
(35)$$E_{r}=E_{\mathrm{OCP}}-E_{\mathrm{PZC}}$$

It is worth emphasizing that adsorption behavior of inhibitors is rarely investigated by these electrochemical techniques, but rather by thermodynamic/kinetic parameters which can be obtained by substituting the main experimental parameters into a series of thermodynamic/kinetic formulas, which will be discussed in more detail below.

### 3.2. Adsorption Isotherms and Thermodynamic Parameters

#### 3.2.1. Adsorption Isotherms

To describe the corrosion inhibitor adsorption behavior or the interaction between metal and inhibitor, adsorption thermodynamics research methods are used by the vast majority of relevant studies. The calculated thermodynamic parameters of adsorption play a major role in understanding the mechanism of the adsorption process of the metal surface. In many cases, it is for parameters such as the adsorption equilibrium constant (K_ads_), adsorption free energy (ΔG_ads_), enthalpy (ΔH_ads_), and entropy (ΔS_ads_) that the adsorption types of an inhibitor on the metal surface (i.e., physisorption or chemisorption, or both) can be distinguished. Before that, different types of adsorption isotherm models, including Langmuir, Temkin, Freundlich, Frumkin, El-Awady, and Flory–Huggins models, need to be evaluated to identify the most fitted model. The isotherm relations and equations for the six adsorption isotherm models are shown in Table 3.

where *θ* is the degree of surface coverage (as determined by Equation (12)), and *C* and *K* display the inhibitor concentration in the corrosion medium and the standard adsorption equilibrium constant, respectively. In equation No.3, 1/*n* is the slope of the linear Freundlich plot (log*θ* vs. log*C*), the value represents the adsorption intensity of tested compound and surface heterogeneity, the value of 1/*n* is between 0 and 1 (0 < 1/*n* < 1), indicating that the adsorption becomes favorable and strong whereas larger than 1 (1/*n* > 1) does the reverse; In equation No.2 and No.4, g is the adsorbate interaction parameter; In equation No.5, y is the number of active sites, if 1/y is less than 1, it indicates multilayer adsorption, and if greater than 1, it represents that a given inhibitor molecule occupies more than one active site; In equation No.6, *n*_FH_ is the Flory–Huggins exponent. 

The above six types of adsorption isotherm models have been widely applied in different corrosion inhibitors, which are primarily suitable for evaluating molecular scale monolayer or multilayer adsorption behavior. For nanoparticles in polar media, several limitations for adsorption processes must be considered, such as the increase in particle size, the difference in excess charges, and even precipitation. In a study on functionalized carbon nanotubes (FCNTs) as corrosion inhibitors of carbon steel, Hongyu Cen et al. cited a three-parameter isotherm model with both Langmuir and Freundlich properties, i.e., the Redlich–Peterson (R–P) equation, which is suitable for heterogeneous and multilayer adsorption systems, as well as for the investigated nanoparticle adsorption behavior [85,86]. The Redlich–Peterson (R–P) equation can be represented by Equation (36).
(36)$$q_{e}=\frac{K_{RP}C_{e}}{1+B_{RP}C_{e}^{\alpha }}$$
where *q_e_* and *C_e_* are the adsorption quantity and equilibrium concentration, respectively; *K_RP_* and *B_RP_* are constants; *α* indicates the heterogeneity of the adsorbent (range 0–1), which can describe the agglomeration effect of nanoparticles on the effective concentration; and an *α* value approaching 1 indicates the uniform adsorption of all nanoparticles.

#### 3.2.2. Thermodynamic parameters

Subsequently, adsorption free energy (Δ*G*_ads_), enthalpy (Δ*H*_ads_), and entropy (Δ*S*_ads_) can be determined by using *K*_ads_ values calculated from the best fit adsorption isotherm. These formulas for the calculation of all the mentioned parameters are given in Equations (37)–(40).
(37)$$\Delta G_{\mathrm{ads}}^{o}=-RT\ln(55.5K_{\mathrm{ads}})$$
where *R* is the universal gas constant, *T* is the absolute temperature, and 55.5 is the molar concentration of water. The Δ*G*°_ads_ value can judge whether the adsorption process is spontaneous or not, and a decreasing tendency for the Δ*G*°_ads_ value refers to a spontaneous process, which is nonspontaneous otherwise [87]. Generally, a Δ*G*°_ads_ value greater than or equal to –20 kJ/mol is related to the electrostatic interaction between charged molecules and the charged metal surface, i.e., physisorption. When the value is less than or equal to –40 kJ/mol, it is interpreted as transferring from the inhibitor molecules to the metal surface to form a coordinate covalent bond or charge sharing, i.e., chemisorption. In the case of –20 kJ/mol ≤ Δ*G*°_ads_ ≤ –40 kJ/mol, indicating the presence of both physical and chemical adsorption [88].

The values of Δ*H*°_ads_ and Δ*S*°_ads_ can be calculated by plotting ln*K*_ads_ vs. 1/*T* straight lines, as the following Van’t Hoff equation (Equation (38)) [34]:(38)$$\ln K_{\mathrm{ads}}=-\frac{\Delta H_{\mathrm{ads}}^{o}}{RT}+\frac{\Delta S_{\mathrm{ads}}^{o}}{R}+\ln\frac{1}{55.5}$$

The equation of the straight line gives the slope and intercept, which are equal to –Δ*H°*_ads_/R and to (Δ*S°*_ads_/R) + ln(1/55.5), respectively. Additionally, the values of Δ*H*°_ads_ and Δ*S*°_ads_ can be calculated by plotting Δ*G*°_ads_ vs. *T* as the following basic thermodynamic equation (Equation (39)), which gives a straight line with slope Δ*S*°_ads_ and intercept Δ*H*°_ads_ [89]. Additionally, the Gibbs–Helmholtz equation can be used to calculate the enthalpy of adsorption, as shown in Equation (40) [34].
(39)$$\Delta G_{\mathrm{ads}}^{o}=\Delta H_{\mathrm{ads}}^{o}-T\Delta S_{\mathrm{ads}}^{o}$$
(40)$$\frac{\Delta G_{\mathrm{ads}}^{o}}{T}=\frac{\Delta H_{\mathrm{ads}}^{o}}{T}+K$$

An endothermic process for the adsorption of inhibitor (Δ*H*°_ads_ > 0) indicates chemical adsorption, while an exothermic process (Δ*H*°_ads_ < 0) suggests that the adsorption of inhibitor may involve physical or chemical adsorption or both, and it can be further differentiated by the absolute value of Δ*H*°_ads_ [90,91,92]. Adsorption of an organic inhibitor at the metal/solution interface can be regarded as the process of water molecule exchange with inhibitor molecules at the corroding interfaces (as mentioned in Equation (12)) [32], which can be represented by a positive Δ*S°*_ads_ value, that is, that the adsorption of one inhibitor molecule over the metal surface leads to the desorption of more water molecules, which in turn causes an increase in disorder [93].

The activation energy (*E*_a_) for a metal in the presence of inhibitors can be calculated by using the Arrhenius equation (Equations (41) and (42)). In most cases, a higher activation energy can be obtained in the presence than in the absence of inhibitor, which is attributed to the formation of an inhibitor–metal complex in solvent, resulting in the higher energy barrier of metal dissolution. In contrast, Radovici [94] proposed that *E*_a_ is smaller in the presence than in the absence of inhibitor when chemisorption occurs. In addition, in a previous study on the role of fatty acids in the adsorption and corrosion inhibition of iron, Szauer and Brand [95] found that a greater adsorption amount of inhibitor molecules due to a higher temperature results in a lower surface area of metal exposure in contact with the solvent environment and, in turn, decreasing corrosion rates with increasing temperature. In this particular case, the authors suggested that this result could be indicated by the decrease in activation energy.
(41)$$\log C_{R}=-\frac{E_{a}}{2.303RT}+\log A$$
(42)$$\log\frac{C_{R}{}_{2}}{C_{R}{}_{1}}=\frac{E_{a}}{2.303R}\frac{T_{2}-T_{1}}{T_{1}T_{2}}$$
where *C*_R_ is the corrosion rate, *A* is the Arrhenius pre-exponential factor, *R* is the universal gas constant, and *C*_R1_ and *C*_R2_ are the corrosion rates at temperature *T*_1_ and temperature *T*_2_, respectively. In some studies, the transition state equation is used to calculate the value of *C*_R_ (as shown in Equation (43)) [34,96,97].
(43)$$C_{R}=\frac{RT}{Nh}\exp(\frac{\Delta S_{a}}{R})\exp(-\frac{\Delta H_{a}}{RT})$$
where *N* is Avogrado’s number, *h* is Plank’s constant, and ∆*S*_a_ and ∆*H*_a_ are the change in enthalpy and change in the entropy of the corrosion process, respectively.

### 3.3. Theoretical Calculations

The values obtained by experimental approaches can reflect average properties, but it is difficult to obtain structural and dynamic information at atomic and molecular scales. With the improvement of computer hardware and software, computational techniques are widely applied in research on corrosion inhibition mechanisms of inhibitor molecules without using environmentally malignant chemicals and lab instruments, which is helpful to establish a connection between molecular structure and inhibition efficiency and to develop new types of corrosion inhibitors. Currently, the theoretical calculation methods for studying inhibitors mainly consist of density functional theory (DFT)-based QM calculations and quantum mechanics/molecular mechanics (QM/MM) MD and MC simulations. 

#### 3.3.1. DFT Calculations

DFT is a promising quantum mechanical approach to provide the accurate structural and electronic properties of compounds. Many global and local reactivity descriptors, defined within the DFT framework, are able to help predict the potential mechanism and corrosion inhibition properties of an inhibitor [98]. In most related studies, some experiments mentioned above are performed first, and then the mechanism of corrosion inhibition is explained at both the molecular structure and microscopic level by calculating descriptors and constructing the prediction models.

(1)Global Reactivity Descriptors

Based on frontier molecular orbital (FMO) theory, only HOMO and LUMO need to be considered when analyzing the chelation process of chemisorption because the chemical reactivity descriptors of most chemical reactions can be calculated by the interaction between the HOMO and/or LUMO of the inhibitor and metal surface. The most used DFT-based quantum mechanical molecular descriptors include the highest occupied molecular orbital energy (E_HOMO_), lowest unoccupied molecular orbital energy (E_LUMO_), energy band gap (ΔE  =  E_LUMO_ − E_HOMO_), ionization potential (*I*), electron affinity (*A*), global hardness (*η*), global softness (*σ*), absolute electrophilicity index (*ω*), global electronegativity (*χ*), dipole moment (*μ*), and the number of electrons transferred (ΔN) [99].

E_HOMO_ reflects the capacity of inhibitor molecules to donate electrons, and E_LUMO_ reflects the capacity of inhibitor molecules to accept electrons. A higher E_HOMO_ value implies that the inhibitor molecule can easily donate electrons to the low energy orbitals, while a lower E_LUMO_ value implies that the metal tends to accept electrons. ΔE between LUMO and HOMO is an important descriptor reflecting the ability of inhibitor molecules to inhibit corrosion reactions. In general, the smaller ΔE is, the lower the energy of electron removal from the last occupied orbit and therefore the higher the inhibition efficiency. Additionally, ΔE is a molecular stability indicator, and a smaller ΔE value implies better stability of the complex formed over the metal surface. The calculated results cited in the literature are shown in Figure 2 [100], from which we can clearly differentiate the magnitude of ΔE value between the three inhibitors to screen out the most potent inhibitor.

From Koopman’s theorem, the orbital energies of the HOMO and LUMO of the inhibitor molecule are related to *I* and *A*, and the relationship of E_HOMO_ versus *I* and E_LUMO_ versus *A* are shown in Equations (44) and (45) [101,102]. Similarly, *I* and *A* indicate the ability of a molecule to donate and accept electrons—the higher the value, the easier the corresponding process can occur. Higher values of *σ*, *χ*, and *ω* or lower values of *η* indicate higher corrosion inhibition. These findings are mainly based on the tendency of inhibitor molecules to accept and donate electrons as well as Pearson’s hard/soft acid/base principle [103,104,105,106].
(44)$$I=-E_{\mathrm{HOMO}}$$
(45)$$A=-E_{\mathrm{LUMO}}$$

*μ* can be used to describe the magnitude of molecular polarity, a higher value of *μ* can be interpreted as the strong interaction between an inhibitor molecule and metal surface, whereas the opposite indicates the accumulation of inhibitor molecules around the electrode surface, but the relationship between it and corrosion inhibition efficiency has not been previously confirmed [107,108].

ΔN obtained based on Pearson’s calculation method represents the electron-donating capacity, if ΔN > 0, the inhibitor molecule passes electrons to the metal, on the contrary retrieves [103]. Earlier studies by Lukovit et al. suggested that the corrosion inhibition effect increases as ΔN increases when ΔN < 3.6, which is due to the increased electron-donating ability of the metal surface [108,109].

The formulas for the calculation of several descriptors mentioned are given in Equations (46)–(50) [103,105,110].
(46)$$\chi =-\frac{E_{\mathrm{LUMO}}+E_{\mathrm{HOMO}}}{2}=\frac{I+A}{2}$$
(47)$$\eta =\frac{E_{\mathrm{LUMO}}-E_{\mathrm{HOMO}}}{2}=\frac{I-A}{2}$$
(48)$$\sigma =\frac{1}{\eta }$$
(49)$$\omega =\frac{\chi^{2}}{2\eta }$$
(50)$$\Delta N=\frac{\chi_{B}-\chi_{C}}{{2(\eta }_{B}+\eta_{C})}$$

(2)Local Reactivity Descriptors

The prediction of inhibitor molecule active sites plays an important role in corrosion inhibition mechanism studies. FMOs, Mulliken charge distribution, Molecular electrostatic potential (MEP), and Fukui functions are generally used to determine the distribution of charge density and possible binding sites. 

As mentioned above, FMOs consist of HOMO and LUMO, and their energies are often associated with their electron donating ability and electron accepting ability. HOMO and LUMO plots (i.e., the distributions of FMOs) of three Schiff-base molecules are adopted from a literature and shown in Figure 3 [111], from which the active site of an inhibitor molecule attacked by electrophile metal cations (HOMO in Figure 3) and the site of an inhibitor molecule to accept electrons (LUMO in Figure 3) can be clearly delineated. 

Mulliken charge distribution can also be used to determine the active sites of the reaction between an inhibitor molecule and a metal surface. In general, the more negative the atomic charge of the adsorption sites is, the easier it would be to donate electrons to nearby unoccupied orbitals. The Mulliken charge distribution is often combined with the HOMO energy density distribution to predict the active site of the inhibitor molecule, i.e., the more negative the atomic charge is and the higher the HOMO energy density is, the more likely the adsorption site is.

MEP refers to the spatial distribution of electrostatic potential generated by nuclei and electrons around molecules, which is a computational tool utilized to predict the chemical reactivity of molecules. The electrostatic potential at any point $\vec{r}$ in the space surrounding a molecule can be expressed as Equation (51) [112].
(51)$$V\left(\vec{r}\right)=\sum_{A}\frac{Z_{A}}{\left|\vec{R_{A}}-\vec{r}\right|}-\int \frac{\rho \left(\vec{r^{\prime }}\right)}{\left|\vec{r}-\vec{r^{\prime }}\right|}dr^{\prime }$$
where *Z_A_* is the charge of the nucleus *A* located at $\vec{R}$ and *ρ*($\vec{r}$) is the molecular electron density function. The sign of V($\vec{r}$) at a particular region depends upon whether the effect of the nucleus or the electrons is dominant there. 

MEP maps are three-dimensional diagrams that can be used to visualize charge distributions and charge-related properties of molecules. They map the contours of the total electron density based on color grading (the different colors represent and visualize different surface electrostatic potentials) from which the active sites of nucleophilics and electrophiles in a molecular system can be determined. In most MEP maps, the red outline corresponds to the largest negative site (easy to donate electrons), and the blue outline corresponds to the largest positive site (easy to accept electrons), which are the preferred locations for electrophilic and nucleophilic attacks, respectively. In a previous study, Sayin et al. determined the most active sites of three aminotriazole inhibitors over carbon steel by using MEP maps, as shown in Figure 4 [113], according to color grading and the environment of nitrogen atoms which is located in the ring and is red than other region. This means that nitrogen atoms located in the rings are more active than other atoms. 

The Fukui function represents the change in the electron density of a molecule at a given position as the number of electrons changes, which is defined as the first derivative of the electron density *ρ*(*r*) with respect to the number of electrons *N* at a constant external potential v(*r*) (as shown in Equation (52)) [114]. A previous study by Koumya et al. [115] demonstrated that a larger *f*(*r*) corresponds to a more reactive active center in a molecule, and the calculation of the Fukui indices ($f_{k}^{+}$, $f_{k}^{-}$ and $f_{k}^{0}$) is helpful to investigate the local reactivity of a molecule. Additionally, Morell et al. first proposed the concept of the dual descriptor (Δ*f*(*k*)), that is, the dual descriptor with a positive sign represents that an atom acts as an electrophilic species, and a negative sign represents that an atom acts as a nucleophilic species [116]. $f_{k}^{+}$, $f_{k}^{-}$, $f_{k}^{0}$ and Δ*f*(*k*) for site *k* in a molecule can be calculated by Equations (53)–(56) [117,118,119].
(52)$$f(r)={\left[\frac{\partial \rho (r)}{\partial N}\right]}_{v(r)}$$
(53)$$f_{k}^{+}=[\rho_{k}(N+1)-\rho_{k}(N)]\;(\mathrm{for}\;\mathrm{nucleophilic}\;\mathrm{attack})\;$$
(54)$$f_{k}^{-}=[\rho_{k}(N)-\rho_{k}(N-1)]\;(\mathrm{for}\;\mathrm{electrophilic}\;\mathrm{attack})\;$$
(55)$$f_{k}^{0}=\frac{1}{2}[\rho_{k}(N+1)-\rho_{k}(N-1)]\;(\mathrm{for}\;\mathrm{radical}\;\mathrm{attack})\;$$
(56)$$\Delta f(k)=f_{k}^{+}-f_{k}^{-}$$
where *ρ*_k_(*N*), *ρ*_k_(*N* + 1), and *ρ*_k_(*N*−1) are the electronic densities of site *k* for neutral, anionic, and cationic species, respectively. The highest values of $f_{k}^{+}$ and $f_{k}^{-}$ are consistent with the most likely sites for nucleophilic and electrophilic attacks, respectively [115].

#### 3.3.2. MD and MC Simulations

MD and MC simulations provide useful information on the adsorption behavior and orientation of corrosion inhibitors on metal–electrolyte interfaces. MD simulations are a computer-based modeling technique by which the evolution as a trajectory of a molecule or function of time can be described by the principles of classical Newtonian mechanics [120]. MC simulations are the first ab initio force field that can be used efficiently, accurately, and simultaneously to predict the condensed-phase properties for a broad range of chemical systems and gas-phase properties, and it is very useful to perform a rapid simulation [32,121,122]. 

In the simulations, the simulation of inhibitor and metal surface interaction is performed in a specifically created simulation box, in which the inhibitor is in direct contact with the metal surface [123]. The simulation results, i.e., adsorption model, consists of the constructed several layers of iron atoms and the optimized structures of the inhibitor [124,125]. Usually, the Fe(110) surface is selected as the adsorption substrate due to a low surface energy, and a great coordination number of substrate atoms causes more active sites of interaction between the inhibitor molecules and the metal [126]. As seen by the cited adsorption model of FMPPDBS compound ((1-(5-fluoro-2-(methylthio) pyrimidine-4-yl) piperidine-4-yl)-2,5-dimethoxybenzenesulfonamide) on different iron surfaces (Figure 5), this fact was well described by Kaya et al. [127], from which the calculated binding energies between Fe(110), Fe(100), and Fe(111) surfaces and FMPPDBS compound were 975.1 for 862.4 and 711.4 KJ·mol^−1^, respectively. It is worth mentioning that a high binding energy leads to a more stable inhibitor/surface interaction, which in turn causes a higher inhibition efficiency.

Moreover, there are many factors to be considered during the simulations, such as the description of van der Waals interactions, the handling of electrostatic interactions, the selection of a temperature control system, the settings of the time step and total simulation time for implementation and the settings of the cut-off distance, spline width, and buffer width [128]. A series of parameters derived from MD and MC simulations are introduced as follows.

The adsorption of inhibitor molecules on metallic surfaces is an exothermic process that results in the evolution of energy in the form of heat. In MC simulations, the total energy (*E*_total_) is defined as the sum of the energies of inhibitor molecules (*E*_inhibitor_), rigid adsorption energy (*E*_rigid_), and deformation energy (*E*_def_) (as Equation (57)) [32].
(57)$$E_{\mathrm{total}}=E_{\mathrm{inhibitor}}+E_{\mathrm{rigid}}+E_{\mathrm{def}}$$
where *E*_rigid_ defines the energy released (or acquired) when the unrelaxed inhibitor molecules adsorb on the metal surface before their geometric optimization. *E*_def_ defines the energy released (or acquired) when inhibitor molecules are relaxed on the metallic surface. The adsorption energy (*E*_ads_) or interaction energy (*E*_int_) is defined as the addition of the above two kinds of energies, which indicates the energy released when one mole of inhibitor molecule adsorbs over the metal surface.

In MD simulations, information regarding the interactions between the molecules and metal surface is determined by the calculation of the interaction energy (*E*_int_) and binding energy (*E*_bin_), as shown in the following relationship (as Equations (58) and (59)) [111,129].
(58)$$E_{\mathrm{int}}=E_{\mathrm{total}}-(E_{\mathrm{surface}+\mathrm{solution}}+E_{\mathrm{molecule}})$$
(59)$$E_{\mathrm{bin}}=-E_{\mathrm{int}}$$
where *E*_total_ represents the total energy of the whole system, *E*_surface + solution_ refers to the total energy of the metal surface and solution without the inhibitor molecule, and *E*_molecule_ denotes the total energy of the inhibitor molecule. A larger *E*_bin_ indicates that the corrosion inhibitor combines with the metal surface more easily and tightly. As seen by the cited adsorption model (Figure 6) [130], the three inhibitor molecules, namely, 3,7-dimethyl-2,6-octadien-1-ol (DTO), santamarine (STA), and lanuginosine (LGS), have different stable adsorption configurations on the Fe (110) surface, while LGS can be adsorbed in a parallel manner, obtaining an anti-corrosion nature that is better than that of DTO and STA, their binding energies on the surface of Fe (110) are 520.2 kJ/mol for DTO, 563.7 kJ/mol for STA, and 756.5 kJ/mol for LGS, respectively.

In order to calculate the potential energy of a system of particles or atoms in a molecular dynamics simulation, a functional form called force fields are used. It is important to choose proper force field in any MD simulation since the force field type has a great impact on the results. Among them, universal and COMPASS (condensed-phase-optimized molecular potentials for atomistic simulation studies) are common force fields which are suitable for simulating organic corrosion inhibitors. Universal is a purely diagonal, harmonic force field. Bond stretching is described by a harmonic term, angle bending by a three-term Fourier cosine expansion, and torsions and inversions by cosine-Fourier expansion terms. Electrostatic interactions are described by atomic monopoles and a screened (distance-dependent) Coulombic term. The van der Waals interactions are described by the Lennard–Jones potential. COMPASS is the first ab initio force field that enables simultaneous and accurate prediction of condensed-phase and gas-phase properties and properties for a broad range of polymers and molecules. COMPASSII is an extension to the COMPASS force field, which extends the existing coverage of COMPASS to include a significantly larger number of compounds of interest to researchers investigating ionic liquids [125].

The chemical bonds of one molecule can bend, stretch, and twist, which leads to a variety of conformations with different potential energies (*E*_pot_). Energy optimization (or the optimized structure mentioned above) during MD simulations is implemented by changing the structures (orientations) of molecules to find the conformation with lower potential energy, and MD simulation results can also exhibit a model with the most stabilized configurations of inhibitor molecule adsorption on the iron surface (the adsorption of molecules on the iron surface in parallel manner can obtain maximum coverage, which is helpful to inhibit metal corrosion). Generally, the potential energy can be represented in terms of a set of molecular coordinates, while the potential energy of the system needs to be represented in the form of a force field, that is, the total potential energy for a collection of molecules with coordinates math is given by the sum of the intramolecular energies of the components plus the sum of the intermolecular interaction energies between all components. Under universal force field, the potential energy of the system is simply expressed as follows (Equation (60)) [131,132].
(60)$$E_{\mathrm{pot}}=E_{\mathrm{bond}}+E_{\mathrm{angel}}+E_{\mathrm{torson}}+E_{\mathrm{charge}}+E_{\mathrm{vander}}+E_{\mathrm{cross}}$$
where *E*_bond_, *E*_angle_, *E*_torsion_, and *E*_charge_ denote the bond length energy, bong angle energy, torsion energy, and energy of charge–charge interactions, and *E*_vander_ denotes van der Waals energy. They constitute the intramolecular potential energy (Equations (61)–(64)) and the intermolecular potential energy (Equation (65)) in the potential energy coordinates of an isolated molecule, respectively [133]. *E*_cross_ denotes miscellaneous energy in the potential energy coordinates of an isolated molecule.
(61)$$E_{\mathrm{bond}}={\sum_{i}k_{b,i}(r_{i}-r_{0,i})}^{2}$$
(62)$$E_{\mathrm{angel}}={\sum_{i}k_{\vartheta ,i}(\vartheta_{i}-\vartheta_{0,i})}^{2}$$
(63)$$E_{\mathrm{torsion}}=\sum_{i}\left[V_{1,i}(1+\cos\phi_{i})/2+V_{2,i}(1+\cos2\phi_{i})/2+V_{3,i}(1+\cos3\phi_{i})/2\right]$$
(64)$$E_{\mathrm{charge}}=\sum_{i<j}\left\{q_{i}q_{j}e^{2}/r_{ij}+4\varepsilon_{ij}\left[{\left(\sigma_{ij}/r_{ij}\right)}^{12}-{\left(\sigma_{ij}/r_{ij}\right)}^{6}\right]\right\}$$
(65)$$E_{\mathrm{vander}}=\sum_{i}^{\in a}\sum_{j}^{\in b}\left\{q_{i}q_{j}e^{2}/r_{ij}+4\varepsilon_{ij}\left[{\left(\sigma_{ij}/r_{ij}\right)}^{12}-{\left(\sigma_{ij}/r_{ij}\right)}^{6}\right]\right\}$$

Unlike the universal force field, COMPASSII force field has more complicated coupling cross terms, as shown in Equation (66) [121,124]. 


(66)
$$\begin{array}{l} E_{\mathrm{pot}}=E_{\mathrm{bond}}+E_{\mathrm{no}-\mathrm{bond}}\\ =\left(E_{b}+E_{\theta }+E_{\varphi }+E_{\chi }+E_{b,b^{\prime }}+E_{b,\theta }+E_{b,\varphi }+E_{\theta ,\varphi }+E_{\theta ,\theta }+E_{\theta ,\theta ,\varphi }\right)+E_{ch\arg e}+E_{vander}\\ =\underset{b}{\sum }\left[k_{2}{\left(b-b_{0}\right)}^{2}+k_{3}{\left(b-b_{0}\right)}^{3}+k_{4}{\left(b-b_{0}\right)}^{4}\right]\\ +\underset{\theta }{\sum }\left[k_{2}{\left(\theta -\theta_{0}\right)}^{2}+k_{3}{\left(\theta -\theta_{3}\right)}^{3}+k_{4}{\left(\theta -\theta_{4}\right)}^{4}\right]\\ +\underset{\varphi }{\sum }\left[k_{1}\left(1-\cos\varphi \right)+k_{2}\left(1-\cos2\varphi \right)+k_{3}\left(1-\cos3\varphi \right)\right]+\underset{\chi }{\sum }k_{2}\chi^{2}\\ +\underset{b,b^{\prime }}{\sum }k\left(b-b_{0}\right)\left(b^{\prime }-{b_{0}}^{\prime }\right)\\ +\underset{b,\theta }{\sum }k\left(b-b_{0}\right)\left(\theta -\theta_{0}\right)+\underset{b,\varphi }{\sum }\left(b-b_{0}\right)\left[k_{1}\cos\varphi +k_{2}\cos2\varphi +k_{3}\cos3\varphi \right]+\underset{b,\theta }{\sum }k\left(\theta -\theta_{0}\right)\left(\theta^{\prime }-{\theta_{0}}^{\prime }\right)\\ +\underset{\theta ,\theta ,\varphi }{\sum }k\left(\theta -\theta_{0}\right)\left(\theta^{\prime }-{\theta_{0}}^{\prime }\right)\cos\varphi +\underset{i,j}{\sum }\frac{q_{i}q_{j}}{r_{ij}}\\ +\underset{i,j}{\sum }\varepsilon_{ij}\left[2{\left(\frac{r_{ij}{}^{0}}{ri_{j}}\right)}^{9}-3{\left(\frac{r_{ij}{}^{0}}{ri_{j}}\right)}^{6}\right] \end{array}$$


The functions of COMPASS force field can be divided into two categories of valence terms, i.e., diagonal and off-diagonal cross-coupling terms and non-bonded interaction terms. These valence terms represent internal coordinates of bond (*b*), angle (*θ*), torsion angle (*φ*), and out-of-plane angle (*χ*), and the cross-coupling terms that include combinations of two or three internal coordinates. Non-bonded interactions include a Coulombic function for an electrostatic interaction and a Lennard–Jones function for the van der Waals term, which are used to indicate the interactions between pairs of atoms that are separated by two or more intervening atoms or those that belong to different molecules. 

In the aqueous phase, the adsorption energy depends on several competing interactions, such as metal–water, inhibitor–metal, and inhibitor–water interactions. Generally, the degree of inhibitor adsorption on the metallic surface in the aqueous phase is adversely affected by salvation [32,128]. The salvation energy (*E*_sal_) is calculated from the following relationship (Equation (67)) [131].
(67)$$E_{\mathrm{sal}}=E_{\mathrm{inh}+\mathrm{water}}-(E_{\mathrm{water}}+E_{\mathrm{inh}})$$
where *E*_inh_ and *E*_water_ are the potential energies of free inhibitor and water, respectively, and *E*_inh + water_ is the potential energy of water–inhibitor interactions.

In addition, the radial distribution function (RDF) g(r) can be further used to extract information about the bond length and the type of interaction of inhibitor molecules on the metal surface from MD simulation data, which is defined as the probability of finding particle B within the range r + dr around particle A. RDF, g(r), is calculated as in Equation (68).
(68)$$g(r)=\frac{1}{{\langle \rho B\rangle }_{\mathrm{local}}}\times \frac{1}{N_{A}}\sum_{i\in A}^{N_{A}}\sum_{j\in B}^{N_{B}}\frac{{\delta (r}_{\mathrm{ij}}-r)}{{4\pi \;r}^{2}}$$
where ‹*ρ*B›_local_ indicates the particle density of B averaged over all shells beside particle A. 

Figure 7 shows the RDF analysis results stemming from the consequence of the MD simulation data of two quinoxaline derivative inhibitor molecules (Q1 and Q2) cited in the literature [134]. These results are shown in the form of a plot of the distance r versus the probability g(r). Generally, the first prominent peak occurs at 1~3.5 Å (an indication of small bond length), implying chemisorption, whereas the peaks with the distance r greater than 3.5 Å are associated with physisorption [135]. According to the results shown in Figure 7a, the bonding lengths of Fe–Q1 and Q2 are 3.10 and 2.90 Å, respectively, suggesting that the two inhibitors have a significant interaction between the two inhibitor molecules and the adsorption sites in the surface of Fe, and the adsorption between them occurred via chemisorption. According to the RDF results shown in Figure 7b,c, the first prominent peaks of nitrogen and oxygen atoms for each type of inhibitor are also less than 3.5 Å, meaning that these atoms are located in the near vicinity of Fe atoms, that is, the interactive force of the oxygen and nitrogen atoms on the Fe surface plays a major role in the adsorption of the two inhibitor molecules.

### 3.4. Physicochemical and Morphology Characterization Techniques

By performing a relevant literature search on corrosion inhibitor, a variety of characterization techniques are found and are listed in Appendix A. Which method belongs to which reference will be noted in Appendix A) of the following sections. Among them, FTIR, UV, Raman, EDS, XPS, SEM, AFM, and TEM are commonly used for characterizing the physicochemical and structural properties of both inhibitors and adsorption layers. Generally, the presence of different functional groups, such as C–N, N–H, C=O, C–H, C=C, O–H, –CN, –CH_2_, and C–S–C, can be confirmed by FTIR, UV, Raman, and XPS. SEM and AFM are commonly used to observe the surface corrosion morphologies, while TEM analysis can characterize the morphology and size of polymeric-nanoparticles corrosion inhibitors. With technological advancements, as well as the demand for the development of multifunctional corrosion inhibitors, numerous other methods are available for investigating the chemical/physical characteristics of inhibitors, such as TGA, CA, NMR, MS, surface tension tests, biodegradability tests, antimicrobial assays, and many more.

## 4. Classification and Statistical Analysis

### 4.1. Drug Molecules

Compounds that act as effective corrosion inhibitors (such as imidazoles, pyridines, thiophenes, furans, isoxazoles, etc.) share substantial similarity with the substructures of many of the commonly used drug molecules. This characteristic has driven scientists to investigate the potential applicability of drugs as corrosion inhibitors.

Golestani et al. [136] studied the effect of penicillin G, ampicillin, and amoxicillin drugs on the corrosion behavior of carbon steel in 1.0 M HCl solution using electrochemical techniques. The results showed that penicillin G exhibited a maximum IE value of 98.4% at a 10 mM concentration at 25 °C. In this study, in addition to potentiodynamic polarization (PDP) and electrochemical impedance spectroscopy (EIS), an electrochemical noise (EN) technique under open circuit conditions was also applied to evaluate the corrosion behavior of these inhibitors, from which the IE value of these inhibitors was also calculated using the standard deviation of partial signal (SDPS) based on the amount of noise charges at a particular frequency interval, and these IE results were similar to those of PDP and EIS. On the other hand, Eddy et al. [137] explained the inhibitory action of penicillin G in terms of its physical adsorption (ΔG_ads_ = 9.65 kJ/mol) on the surface of mild steel in 2.5 M H_2_SO_4_ solution. They proposed that the adsorption of penicillins on mild steel surfaces is controlled by weak intermolecular interactions. However, Alder et al. [138] confirmed that penicillins are especially influenced by pH due to the fast degradation of the chemically unstable *β*-lactam ring, so Eddy’s results cannot be generalized in other pH solutions.

Farahati et al. [139] investigated and compared the inhibition performance between D-penicillamine (a degradation product of penicillin) and L-cysteine (an amino acid commonly found in living organisms) on mild steel corrosion in 1.0 M HCl solution by electrochemical methods. These experiments were performed at various concentrations, different immersion times, and different temperatures. The calculated maximum IE value of 5 mM L-cysteine in 1.0 M HCl solutions at 4 h immersion times was 91% through a polarization study. The results also showed that the two drugs acted as mixed-type inhibitors at different concentrations, and their adsorption on mild steel surfaces obeyed the Langmuir isotherm. Figure 8 illustrates SEM images of polished mild steel surface, and its corrosion surface that was soaked in 1.0 M HCl with and without inhibitors (5 mM L-cysteine and D-penicillamine). After 4 h, the damaged and pitted sample surface was clearly observed in the absence of an inhibitor (Figure 8b), while only a few holes or pits were observed due to the formation of a protective layer on the steel surface in the presence of 5 mM L-cysteine (Figure 8c) or D-penicillamine (Figure 8d), where pits were not obvious relatively in the presence of L-cysteine, indicating better corrosion inhibition of L-cysteine compared to that of D-penicillamine. Moreover, the study also noted that D-penicillamine is structurally similar to L-cysteine, and only the two methyl groups in D-penicillamine displace the two H atoms of L-cysteine, the lower inhibiting effect of D-penicillamine compared to L-cysteine being related to the steric effect of the methyl group on D-penicillamine. The same reason could explain the results of the study comparing the inhibition efficiency of L-cysteine and S-methyl cysteine obtained by Amin et al. [140]. In addition, L-cysteine has also been extensively applied as a corrosion inhibitor for a variety of metals and alloys, such as copper [141,142].

The inhibition effect of fluconazole and its fragments as corrosion inhibitors of API 5L X52 steel in 1.0 M HCl solution was studied by Espinoza-Vázquez et al. [143]. EIS measurements suggested that fluconazole, as an active substance at low concentrations, displayed good corrosion inhibition (IE > 80%), and the inhibition efficiency at a 30 ppm concentration (24 h) was optimal. According to thermodynamic analysis, the fluconazole studied followed a physisorption-chemisorption mechanism, while its fragments followed a physisorption mechanism. In another system (aluminum in 0.1 M HCl solution), the inhibitive action of fluconazole was also investigated by Obot et al. [144,145]. They suggested that the adsorption of protonated fluconazole to a positively charged aluminum surface in 0.1 M HCl could occur through oxygen of the hydroxyl group, nitrogens of the triazole rings and fluoride on the benzene ring. The inhibition effect of pheniramine on mild steel corrosion in 1.0 M HCl was investigated by Ishtiaque Ahamad et al. [146] using several techniques, such as weight loss measurements, electrochemical measurements, and surface analyses. The results showed that pheniramine acted as a mixed-type inhibitor and exhibited a maximum efficiency of 98.1% at 0.833 mM and 308 K through weight loss measurements, and the adsorption of pheniramine on a mild steel surface obeyed the Langmuir adsorption isotherm. Similarly, the adsorption of pheniramine at the mild steel/acid solution interface was supported by SEM analyses. Pheniramine is a class of histamines that are similar to meclizine and famotidine, and the inhibition mechanisms of the two drugs previously reported showed great similarity with that of pheniramine [147,148]. The inhibition effect of streptomycin on mild steel corrosion in 1.0 M HCl using weight loss and electrochemical methods was investigated by Shukla et al. [149]. Weight loss studies showed a maximum inhibition efficiency of 88.5% at a 500-ppm concentration of streptomycin. Adsorption of the drug on a mild steel surface followed the Langmuir isotherm, and a polarization study proved that the drug acted as a mixed-type inhibitor. The inhibition effect of streptomycin is attributed to the large molecular size and the existence of π electrons and quaternary nitrogen atoms. In an acidic solution, streptomycin exists as a protonated species. These protonated species are adsorbed on the cathodic sites of mild steel surfaces, while π-electrons of aromatic rings and lone pairs of electrons of nitrogen atoms are adsorbed on the anodic sites. Therefore, they decrease the evolution of hydrogen and anodic dissolution of mild steel, respectively [150]. In acidic solutions, the most favorable stability conditions for streptomycin are below 28 °C and between pH 3 and 7, and the constraints can slow chemical degradation into streptidine and streptobiosamine [151,152]. The inhibition effect of atenolol on mild steel corrosion in 1.0 M HCl was studied by Karthik et al. [153] using several experimental techniques. In the case of potentiodynamic polarization, atenolol exhibited a maximum efficiency of 93.8% at a 300-ppm concentration and behaved as a mixed-type inhibitor. Adsorption of atenolol on the steel surface obeyed the Langmuir adsorption isotherm. Atenolol is easily available, environmentally friendly, and nontoxic, and has a large molecular weight (266.336 g/mol), which is likely to effectively cover more of the surface area of mild steel. Alkanolamine and aromatic groups present in atenolol determine its physicochemical properties, in which the inhibition efficiency in acidic environments is determined by the aromatic ring moiety of atenolol. This view has also been confirmed in previous studies by Fouda et al. [154] on the mechanism of atenolol’s corrosion inhibition of aluminum in 0.1 M HCl solution. Expired tramadol’s corrosion inhibitor effect for mild steel in 1.0 M HCl solution was also tested by Dohare et al. [155] using several techniques, such as weight loss measurements, electrochemical measurements, surface analysis, and density functional theory (DFT) methods. The studied drug showed a maximum inhibition efficiency of 97.2% at 100 mg·L^−1^, exhibited mixed-type behavior and predominantly suppressed the cathodic process, and its adsorption followed the Langmuir isotherm, while also theoretically estimating the structural aspects of neutral and protonated tramadol on the surface of mild steel (by DFT analysis). Prabhu et al. [156] reported that tramadol at a concentration of 21.6 × 10^−4^ M had a higher corrosion inhibition efficiency for mild steel in HCl (82.6% IE) than that in H_2_SO_4_ (76% IE) because tramadol molecules with cationic properties were more easily adsorbed on the surface of steel in HCl solution than that in H_2_SO_4_ solution.

Sulfonamides (sulfa drugs containing the sulfanilamide molecular structure) refer to any compound that contains a SO_2_NH_2_ moiety, and they are one of the oldest classes of antimicrobial compounds. The molecular size, the type of substituent group, and the functional adsorption atoms of the sulfa drug molecule play important roles in the corrosion inhibition process [157]. Samide et al. [158] found that in the presence of sulfacetamide, the corrosion of carbon steel in 1.0 M HCl solution was slowed, and the inhibition efficiency of sulfacetamide reached 84.7% at a concentration of 10 mM. El-Naggar [157] compared the corrosion inhibition effects of sulfaguanidine, sulfadimethazine, sulfadimethoxazole, and sulfadiazine on mild steel corrosion in 1.0 M HCl solutions using galvanostatic polarization and weight loss techniques. The results showed that all these compounds behave as a mixed inhibitor type with predominant cathodic effectiveness, and sulfadiazine shows the maximum corrosion inhibition efficiency among these sulfa drugs. In addition, the inhibition efficiency of sulfa drug compounds can be influenced by the anionic nature of solutions in acidic solutions. For example, sulfa drug compounds show better inhibition performance in HCl solutions than H_2_SO_4_ solutions, mainly due to a smaller degree of hydration for the specific adsorption of Cl^−^.

Cefalexin is a zwitterion containing both basic and acidic groups, and its hydrophobic properties vary with the solution pH. Generally, cefalexin mainly exists in the form of zwitterionics in neutral solutions and cations in acidic solutions [159]. Cefalexin is one of the most commonly used cephalosporins as corrosion inhibitors for carbon steel in acidic media [160]. Shukla and Quraishi [161] studied the corrosion inhibition of cefalexin for mild steel in 1 N HCl using electrochemical and weight loss measurements. The inhibition efficiency of cefalexin increased to 92% as the inhibitor concentration increased to the optimum concentration (400 ppm). The inhibition efficiency was found to decrease with increasing temperature in the range of 35~65 °C. The results also indicated that the adsorption of protonated cefalexin molecules on mild steel is physical adsorption caused by electrostatic attractions.

Analgin is an analgesic drug (also known as Metamizole or Dipyrone), which is almost non-toxic for humans and environment. Bashir et al. [162] investigated the corrosion inhibition property of analgin on mild steel in 1.0 M HCl using weight loss and electrochemical experiments, in which weight loss studies showed that analgin drugs obtained a maximum inhibition efficiency of 96.1% at 4000 ppm and 298 K. Adsorption of the drug on mild steel surfaces followed the Langmuir isotherm. PDP study showed that atenolol acts as a mixed-type inhibitor. Theoretical and experimental results suggested that the inhibition efficiency of an inhibitor is largely dependent on the presence of heteroatoms.

In recent years, drugs as corrosion inhibitors for carbon steel in HCl solutions also include moxifloxacin, antihypertensive drugs losartan, irbesartan, venlafaxine, fexofenadine, pioglitazone, omeprazole, third-generation cephalosporins (ceftriaxone and cefotaxime), etc. It should be noted that not all drug inhibitors have the ability to biodegrade, and their transformation products may be hazardous to the environment, and therefore the nontoxic properties of these drugs must be kept in mind as a critical prerequisite. Appendix A summarizes the techniques, methods, instruments used, and the results obtained in previous research regarding drugs as corrosion inhibitors for carbon steel in 1.0 M HCl media [67,148,163,164,165,166,167,168,169].

### 4.2. Ionic Liquids

Ionic liquids (ILs) deliver an unlimited number of potential derivatives with a wide range of physiochemical properties, and therefore IL compounds can also be used as sustainable and environmentally friendly corrosion inhibitors for metals in different corrosive environments. Compared to highly volatile traditional toxic corrosion inhibitors, ILs have several favorable physicochemical properties, including high solubility (highly soluble in polar corrosive environments due to their ionic nature), low toxicity, less volatility (readily regenerated and reusable, and little environmental pollution), nonflammability, high ionic conductivity, and high thermal and chemical stability. Several types of synthetic ionic liquids as effective corrosion inhibitors for carbon steels in 1.0 M HCl solution are presented in the following paragraphs.

Imidazole-based ionic liquid derivatives are a popular class of compounds as corrosion inhibitors, in which the N=C–N region in the imidazole ring is the active site for the adsorption process of imidazole inhibitor compounds. Mashuga et al. [170] investigated the corrosion inhibition properties of four 1-hexyl-3-methylimidazolium-based ionic liquids (namely, 1-hexyl-3-methylimidazolium trifluoromethanesulfonate, 1-hexyl-3-methylimidazolium tetrafluoroburate, 1-hexyl-3-methylimidazolium iodide, and 1-hexyl-3-methylimidazolium hexafluorophosphate) for mild steel in 1.0 M HCl. Their study showed that the four studied ionic liquids acted as good corrosion inhibitors for mild steel in 1.0 M HCl, and their IE at 500 ppm are 79.88%, 78.54%, 71.55%, and 79.81%, respectively. The adsorption mechanism of these IL molecules on mild steel surfaces might occur via physical and chemical adsorption. Electrochemical results revealed that these ionic liquids behaved as mixed-type inhibitors. The different inhibitory capacities of these ILs studied might be ascribed to the different anions in these 1-hexyl-3-methylimidazolium-based ionic liquids, which results in their different critical aggregation concentration (CAC) values. Guo et al. [171] synthesized two ionic liquids, namely, 1-vinyl-3-aminopropylimidazolium hexafluorophosphate ([VAIM][PF6]) and 1-vinyl-3-aminopropylimidazolium tetrafluoroborate ([VAIM][BF4]), using a two-step synthesis according to the literature procedure [172,173], and studied their inhibition performance on carbon steel corrosion in 1.0 M HCl using several experimental techniques. Weight loss measurements showed that the corrosion inhibition efficiency for [VAIM][PF6] and [VAIM][BF4] were 90.53% and 54.01% at 45 °C, and their adsorption types over the carbon steel surface followed the Langmuir adsorption isotherm and EI-Awady kinetic-thermodynamic adsorption isotherm, respectively. UV–vis spectroscopic measurement results supported the formation of the Fe^2+^-IL complex on the carbon steel surface. The authors also proposed a new point to explain the higher corrosion inhibition efficiency of ILs with hydrophobic anions than that with hydrophilic anions on the basis of results obtained from mechanism analyses. On the other hand, Sasikumar et al. [174] pointed out that the corrosion inhibition efficiency of 1-ethyl-3-methylimidazolium tetrafluoroborate, 1-butyl-2,3-dimethylimidazolium tetrafluoroborate and 1-decyl-3-methylimidazolium tetrafluoroborate was affected by the length of the alkyl side chain through experimental, QC, and MC simulation studies.

Ammonium-based ionic liquids are another class of IL compounds that have been investigated extensively as corrosion inhibitors for carbon steel. In a study, Kowsari et al. [64] synthesized an ammonium-based ionic liquid, namely, tetra-n-butyl ammonium methioninate, and the role of this inhibitor in corrosion protection of mild steel exposed to 1.0 M HCl was reported. They observed that the adsorption of this IL compound on the surface followed the Frundlich isotherm and that the adsorption process occurred through electrostatic interactions between the inhibitor and iron (physical adsorption). In another study, the corrosion inhibition behavior for three different ammonium-based IL compounds (namely, 2-hydroxyethyl-trimethyl-ammonium cations with three different anions chloride, iodide, and acetate) on mild steel in 1.0 M HCl were compared by Verma et al. [41] using weight loss experiments, PDP, EIS, DFT calculations, and MC simulations. The results suggested that these ILs acted as mixed-type corrosion inhibitors, their adsorption mode obeyed the Temkin adsorption isotherm, and their inhibition efficiencies achieved maximum values at 17.91 × 10^−4^ M, which were 92.04%, 96.02%, and 96.59%, respectively. 

Ionic liquids based on pyrrolidine are also used as corrosion inhibitors in corrosive solutions. Pyrrolidinium compounds contain a saturated five-membered ring that is joined by four carbon atoms and one positive nitrogen atom, and the inhibition process is mainly achieved by their adsorption on the metal surface. The inhibition efficiencies of four ionic liquids based on pyrrolidine with different anions and cations on mild steel corrosion in 1.0 M HCl were compared by Al-Rashed et al. [175] using PDP, EIS, SEM, and FTIR. The experimental data indicated that the ILs with –N(CN)_2_ anions had higher corrosion inhibition efficiency than those with –SCN and –PF6 anions, while the increase in the length of alkyl chains on the cations led to an increase in the protection of mild steel from corrosion. With the same anion species and alkyl chain length, the ILs containing imidazolium cations had higher corrosion inhibition efficiency than those containing pyrrolidinium and pyridinium cations. Among these, pyridinium is a class of heterocyclic cationic compounds that contain a six-membered ring with five carbon atoms and one nitrogen atom, and multiple ILs based on pyridinium have been considered as corrosion inhibitors of carbon steel in 1.0 M HCl solution. For example, one ionic liquid based on pyridinium (1-(3-bromopropyl)-4-(dimethylamino)pyridiniumbromide) was synthesized and characterized (using ^1^H NMR and ^13^C NMR and DEPT spectroscopies) by Aoun et al. [176]. Next, they studied the adsorption and inhibition effect of the IL inhibitor on carbon steel in a 1.0 M HCl solution by WL, PDP, and EIS. These results indicated that the IL inhibitor has mixed-type inhibition properties, its physical adsorption process on the carbon steel surface obeyed Langmuir’s adsorption isotherm, and the maximum corrosion inhibition efficiency of 90% was obtained at a 3 mM concentration. In another work, Aoun et al. [177] also analyzed the inhibition performance of another ionic liquid based on pyridinium, namely, 1-hexylpyridinium bromide on carbon steel, in 1.0 M HCl by WL, electrochemical measurements, thermodynamics studies, and SEM studies. The study findings suggested that the investigated IL inhibitor was considered as a mixed-type inhibitor by LPR measurements, the maximum inhibition efficiency of 88.6% was obtained at a 3 × 10^−3^ M concentration and 294 K, and the adsorption of the IL inhibitor molecules on the carbon steel surfaces obeyed the Langmuir adsorption isotherm model with a predominant physisorption nature.

Pyridazium-based ILs also showed excellent inhibition performance due to the existence of π-electrons and N and S heteroatoms. Bousskri et al. [178] investigated the corrosion inhibition performance of 1-(2-(4-chlorophenyl)-2-oxoethyl) pyridazinium bromide on carbon steel in 1.0 M HCl solution at different concentrations and temperatures. The results showed that the inhibition efficiency of the studied IL inhibitor decreased slightly with increasing temperature and reached the highest value (91%) at 10^−3^ M and 298 K. Messali et al. [179] synthesized a pyridazinium-based ionic liquid, i.e., 1-(2-(4-Nitrophenyl)-2-oxoethyl) pyridazinium bromide, and then investigated the IL as a corrosion inhibitor for carbon steel in 1.0 M HCl by using WL, electrochemical tests, and surface studies. They observed that the corrosion inhibitory performance of the inhibitor increased with the addition of concentration, reaching a maximum value of 88% efficacity at 10^−3^ M and 328 K, which could be a suitable inhibitor for high-temperature environments. Additionally, the IL compound behaved as a mixed inhibitor. Its adsorption obeyed the Langmuir adsorption isotherm model, confirming the chemisorptive nature of the IL inhibitor molecule adsorption on carbon steel based on thermodynamic analysis. 

From the above literatures, several classes of IL-based compounds mentioned can be used as effective and ecofriendly corrosion inhibitors for carbon steel in 1.0 M HCl solution, in which imidazole-based ionic liquids have been used most extensively. Benzyltributylammonium tetrachloroferrate ionic liquid synthesized by Kannan et al. [180] proved 99.5% inhibition efficiency (at 300 ppm concentration) for carbon steel in 1.0 M HCl solution, which showed the highest corrosion inhibition efficiency in this part of the investigation. It is worth mentioning that some IL compounds with amino acid groups in the anion structure are defined as amino acid-based ionic liquids, and amino acids are cheap, readily available, and ecofriendly compounds with good solubility in aqueous media, thus, many of these IL compounds are also used as corrosion inhibitors in 1.0 M HCl media for carbon steel [171,181,182,183]. In addition, the corrosion inhibition behavior of many other types of IL compounds has also been investigated in different metal/electrolytic liquid systems, such as IL inhibitors based on triazolium, thiazolium, indolium, picolinium, piperidinium, thiazinium, phosphonium, chitosan, acrylamide, sulfonium, and so on. These findings provide best-practice recommendations and new directions for IL inhibitor development for carbon steel in HCl solutions. Appendix A summarizes the techniques, methods, and instruments used, and the results obtained in previous research regarding ILs as corrosion inhibitors for carbon steel in 1.0 M HCl media [66,183,184,185,186,187,188,189,190,191].

### 4.3. Surfactants

Surfactants and their mixtures have been widely used as corrosion inhibitors for the protection of metallic materials in various solution environments, which can drastically change the properties of metal/metal oxide–water interfaces due to their amphiphilic nature, that is, that the hydrophilic (polar) group of the surfactant attaches to the metal surface, and the hydrophobic (nonpolar) moiety extends away from the interface toward the solution to form an array of hydrophobic tails [192,193,194].

Hegazy et al. [195] synthesized three nonionic surfactants I, II, and III (dodecyl 14-hydroxy-3,6,9,12-tetraoxatetradecyl phthalate (compound I), dodecyl 26-hydroxy-3,6,9,12,15,18,21,24-octaoxahexacosyl phthalate (compound II), and dodecyl 41-hydroxy-3,6,9,12,15,18,21,24,27,30,33,36,39-tridecaoxahentetracontyl phthalate (compound III)) and evaluated their inhibition effect on carbon steel in 1.0 M HCl solution. The results showed that the inhibition efficiency of the studied nonionic surfactants followed the order III > II > I, and a good synergistic effect occurred with the addition of inhibitors and KI (10^−3^ M of inhibitor III + 10^−3^ M of KI) in the carbon steel/1.0 M HCl system. In another study, three nonionic surfactants based on azodye and Schiff base condensed with polyethylene glycol were prepared by Bedair et al. [196]. Their maximum inhibition efficiencies at 7.5 × 10^−4^ M were all greater than 92% in the carbon steel/1.0 M HCl systems. Additionally, the authors found that their corrosion inhibition efficiency increased with decreasing surface tension and critical micelle concentration (CMC). 

For another new cationic surfactant based on tolyltriazole, Hegazy et al. [197] synthesized 1-dodecyl-methyl-1H-benzo[d][1%#x2013;3]triazole-1-ium bromide (1-DMBT), which was characterized by using ^1^H NMR, ^13^C NMR, and FTIR spectroscopy, and then investigated its inhibitive action on carbon steel in 1.0 M HCl solution. The results revealed that its IE value increases with increasing concentration but decreases with increasing temperature, where the maximum IE value at 10^−3^ M and 298 K was found to be 97.3%. Furthermore, it was shown that the inhibition mode of the inhibitor occurs via physical and chemical adsorption on the steel surface, obeying the Langmuir isotherm model. In another work, Badawi et al. [198] prepared three cationic surfactant-based alkyl dimethylisopropylammonium hydroxides symbolized by refluxing decyl, dodecyl, and hexadecyl bromides with one mole of dimethylisopropylamine (DEDIAOH, DODIAOH, and HEDIAOH), and then compared the inhibition performance of these cationic surfactants as corrosion inhibitors for carbon steel in 1.0 M HCl. The authors found that the surface excess concentration decreased with increasing carbon chain length due to the hydrophobic effect of the carbon chain. The maximum surface excess (*Γ*_max_) was 1.50 × 10^10^, 1.32 × 10^10^, and 1.19 × 10^10^ mol·cm^−2^ at 30 °C for DEDIAOH, DODIAOH, and HEDIAOH, respectively. Thus, the minimal area occupied by per surfactant molecule adsorbed (*A*_min_ (Å^2^), as estimated from Equation (69) below), would increase in the following order: DEDIAOH < DODIAOH < HEDIAOH. According to gravimetric and polarization measurements, authors also showed that their inhibition efficiency increased with concentration, HEDIAOH had a higher inhibiting effect than DEDIAOH and DODIAOH, and its maximum value of inhibition efficiency was 96.8% after 24 h at 1 × 10^−2^ M. Meanwhile, the prepared cationic surfactants are also considered to be a good surfactant biocide against.
(69)$$A_{\min}=\frac{10^{16}}{\Gamma_{\max}N_{A}}$$
where *N*_A_ is the Avogadro’s number (6.02 × 10^23^ mol^−1^).

In another work, three anionic chitosan surfactants with different hydrophobic tails, designated as chitosan-R8, chitosan-R12, and chitosan-R16, were prepared by Badr et al. [199]. Their inhibition efficiencies for carbon steel in 1.0 M HCl solution followed the order Chitosan-R8 < Chitosan-R12 < Chitosan-R16 (chitosan-R16 surfactant displayed a maximum efficiency of 92.6% at a concentration of 800 ppm). Furthermore, it was shown that both an increase in the hydrophobic chain length and an increase in the solution temperature (20~60 °C) were found to cause a significant increase in the affinity of these anionic surfactants to form micelles. 

For another subject of intense interest, a surfactant molecule consists of two typical surfactant monomers with identical or nonidentical hydrophobic tails covalently linked together by either a rigid or flexible spacer group, either a short chain or long chain spacer group or a polar or nonpolar spacer group, which is called a Gemini surfactant (Figure 9) [200,201,202,203]. Gemini surfactants show a better effectiveness than their corresponding monomer counterparts in micelle formation and in reducing solution surface tension [204,205]. In a recent study, imidazolium Gemini surfactant ([C_14_-4-C_14_im]Br_2_) and its corresponding monomer ([C_14_mim]Br) were synthesized by Zhou et al. to compare their inhibition performance for A3 carbon steel corrosion in 1.0 M HCl solution [206]. The results suggested that the inhibition efficiency of [C_14_-4-C_14_im]Br_2_ was higher than that of [C_14_mim]Br at the same concentration, which may be attributed to the lower CMC value and the tighter adsorption layer for [C_14_-4-C_14_im]Br_2_. In a similar study, Motamedi et al. found that butanediyl-1,4-bis(dodecyldimethylammonium) bromide (12-4-12) has a stronger tendency to self-assemble than its corresponding monomeric counterpart, which results in a higher surface activity and lower CMC value of 12-4-12 [207]. Similarly, Aslam et al. [208] synthesized and characterized oppositely charged biodegradable cationic di-ester bonded gemini surfactants (ethane-1,2-diylbis(N,N-dimethyl-N-alkylammoniumacetoxy)dichloride with 12, 14 and 16 carbon atoms in the alkyl chain, referred to as 12-E2-12, 14-E2-14, 16-E2-16), and then the inhibition performance of mixed solutions of various concentrations (10–700 ppm) of sodium salt of carboxymethylcellulose (NaCMC) and fixed concentrations (1 ppm) of synthesized surfactants were investigated by various experiments and theoretical calculation methods. The results showed that the corrosion inhibition performance of the polymer + surfactant system (500 + 1 ppm) was found to be a good mixed-type inhibitor for mild steel corrosion in 1.0 M HCl solution at 30 °C, the maximum IE of the NaCMC/16-E2-16 system was 90.1% at 30 °C according to the weight loss measurements, while the IE of NaCMC was only 57.3%. In another work, three Schiff base-based cationic Gemini surfactants with different chain lengths were synthesized and characterized by Hegazy [209], and then the feasibility of these surfactants as corrosion inhibitors for carbon steel in 1.0 M HCl solution was further evaluated. The authors found that the inhibition efficiency of 14-S-14 (97.75%) was better than that of both 10-S-10 and 12-S-12 (96.06% and 94.58%) at 25 °C and 5 × 10^−3^ M. 

By collating information from the literature, we found that the majority of these surfactants, as corrosion inhibitors, have shown a high corrosion inhibition efficiency exceeding 90%. Among them, single surfactants (nonionic surfactants, cationic surfactants, anionic surfactants, and Gemini surfactants) as corrosion inhibitors are more common in the literature. Mixtures of various types (anionic−cationic, anionic−nonionic, cationic−nonionic, and nonionic−nonionic) as well as surfactants that cooperate with other organic compounds or inorganic salts as corrosion inhibitors are also mentioned in a few studies. Many ionic surfactant inhibitors contain longer hydrophobic alkyl chains (more than 7 carbons) than ionic liquid inhibitors, which is a significant difference between them. Notably, nonionic surfactants are considered less toxic than ionic surfactants [210]. Additionally, an increase in the chain length of the surfactant molecule leads to a decrease in hydration (hydrophilic part), an increase in the tendency to form micelles, and a decrease in the CMC value of the surfactant [211], but terminal chains that are too long (exceeding 14 carbon atoms) may reduce the solubility of surfactants, which reduces their corrosion inhibition efficiency. Appendix A summarizes the techniques, methods, instruments used, and the results obtained in previous research regarding surfactants as corrosion inhibitors for carbon steel in 1.0 M HCl media.

### 4.4. Plant Extracts

Plant extracts contain complex phytochemicals, and some of these phytochemicals display the same molecular and electronic structures as many organic corrosion inhibitors. In recent years, a number of reports have been published to describe plant extracts’ corrosion inhibition effects for metals in various electrolytic systems, in which most of them exhibit effective corrosion inhibition performance. Meanwhile, the characteristics of abundant feedstock, environmental friendliness, and easy separation are also the most common reasons for investigating plant extracts as corrosion inhibitors.

Magnolia grandiflora leaf contains four main chemical constituents, namely, 3,7-dimethyl-2,6-octadien-1-ol (DTO), santamarine (STA), lanuginosine (LGS), and anonaine (ANI). In a study by Chen et al. [130], Magnolia grandiflora leaf extract was obtained by using the water extraction method, and then the corrosion inhibition performance of the extract was evaluated for Q235 steel corrosion in 1.0 M HCl using different methods. The results showed that the extract behaved as a mixed-type corrosion inhibitor, and its corrosion inhibition efficiency exceeded 85% at 298 K and 500 mg/L, in which LGS provided the maximum contribution to corrosion protection, its adsorption on the Q235 steel surface obeyed the Langmuir adsorption isotherm and followed a physiochemistry mechanism. In another study, the corrosion inhibition property of an aqueous extract of Aloysia citrodora leaves for mild steel in 1.0 M HCl was reported by Dehghani et al. [212] using morphological, electrochemical, and theoretical studies. AFM, SEM micrographs, and the contact angle test results suggested that the surface roughness degree and the surface hydrophilicity index decreased with increasing extract content. PDP and EIS studies showed that the extract behaved as a mixed-type corrosion inhibitor, and the presence of the extract increased the value of total resistance and obtained the highest inhibition efficiency of 94% (at 800 ppm after 2.5 h). Moreover, the chemical adsorption of the extract on a mild steel surface was evidenced by MC, MD, and DFT simulation studies. In another study by the same authors [213], the corrosion protection capability of Laurus nobilis leaf extract was studied in the same metal/electrolytic system, the schematic preparation process of Laurus nobilis leaf extract powder in this study is shown in Figure 10 and Figure 11 demonstrates the geometry of neutrally charged elemicin, eugenol, limonene, santamarine, spatulenol, and terpinyl acetate compounds found in Laurus nobilis leaf extract. The results showed that in the mild steel/1.0 M HCl system, the extract had the highest protection performance of 95.7% at a 400 ppm concentration after 2.5 h. DFT studies suggested that the studied extract acted as an effective adsorbent over the carbon steel surface, which became effective by donor–acceptor interactions. Additionally, Wang et al. [214] proved the possibility of Ficus tikoua leaf extract as a corrosion inhibitor for carbon steel in 1.0 M HCl. The authors identified that 5-methoxypsoralen (5-MOP) components in the extract played the most important role in steel corrosion inhibition by quantum chemical calculations. At 200 mg/L, this extract exhibited excellent protection performance in the temperature range from 298 K to 318 K, which were 95.8% (298 K), 95.7% (308 K), and 95.0% (318 K), respectively.

In a recent study, plant-based corrosion inhibitors of Papaver somniferum leaves/stems extract (PSLSE) for mild steel in 1.0 M HCl were investigated by Majd et al. via various experimental methods [215]. The inhibition mechanism of the extract showed a mixed type of performance with a dominant action on the cathode. EIS studies suggested that their corrosion inhibition efficiencies were 94.7%, 95.5%, 96.33%, 95%, and 96.4% after immersing mild steel into 1.0 M HCl solution in the presence of different concentrations of PSLSE for 5 h (200, 400, 600, 800, and 1000 ppm), while the optimum inhibition efficiency in this study was 97.64% at 600 ppm after 11 h of immersion, this indicated that secondary inhibition (the formation of protective complexes, 11 h) had higher performance than primary inhibition (the adsorption of inhibitors on the active sites of the surface, 5 h), which could be ascribed to the hydrophilic property of protective complexes. Moreover, many effective functional groups on the surface of adsorption films were detected, such as Fe–C, Fe–N, and Fe–O bonds identified by FTIR, iron cyanide, and a supermolecule identified by GI-XRD, and C=C, C=O, and O–H bonds identified by UV–vis, the cited analysis results are depicted as follows (Figure 12).

In other similar studies, the ethanolic extract of Tunbergia fragrans was prepared by Muthukumarasamy et al. [216], and then the inhibition action of the extract for mild steel corrosion in 1.0 M HCl was studied using WL, PDP, and EIS. They observed that the extract acted as a mixed-type corrosion inhibitor, and the inhibition efficiency was enhanced with an increase in inhibitor concentration to reach a maximum of 81.1% at 500 ppm. Moreover, EIS studies showed that phytoconstituents of the extract could effectively adsorb on mild steel surfaces and block corrosion sites. Ji et al. [217] compared the corrosion inhibition potential (in a mild steel/1.0 M HCl system) of aqueous extracts of Musa paradisica peels with different maturity stages, namely, raw M. paradisica peel extract (RMPPE), ripe M. paradisica peel extract (RIMPPE), and over ripe M. paradisica peel extract (ORIMPPE). The results showed that the inhibition efficiency of RMPPE reached 90% at 300 mg/L and 26 °C, which achieved the maximum corrosion inhibition among the studied extracts, followed by ORIMPPE and RIMPPE. Moreover, the authors found that the high contents of gallocatechin and catechin in these extracts were effective components for corrosion inhibition by HPLC studies. Loganayagi et al. [218] conducted a study on Opuntia elatior fruit extract as a corrosion inhibitor for mild steel in 1.0 M HCl utilizing weight loss and electrochemical measurements at temperatures of 303, 313, and 323 K. Resluts revealed the adsorption mechanism between mild steel and Opuntia elatior fruit extract was best fitted to the Temkin adsorption isotherm, the increase in IE values with the increase in the extract concentration, and the highest efficiency of approximately 88% was noted at 500 ppm at 303 K. Moreover, the corrosion inhibition effects of the extract can be related to the presence of opuntiol, while the synergistic effects of opuntiol with various phytoconstituents (such as proline, linolenic acid, campesterol, and betacyanin) can lead to more efficient corrosion inhibition for mild steel in 1.0 M HCl. Bahlakeh et al. [219] explored the inhibiting impact of Mustard seed extract on the corrosion of mild steel in 1.0 HCl at 25, 35, 45, and 55 °C using WL and electrochemical techniques. WL and electrochemical studies showed maximum efficiencies of 97% and 94%, respectively, at 25 °C and 200 mg/L. Additionally, several adsorption isotherms were described, in which the Langmuir adsorption isotherm best explained the adsorption mechanism.

A number of studies have suggested that plant extracts acquired from various portions of plants (including leaves, stems, flowers, peels, fruits and seeds, etc.) are used as corrosion inhibitors widely, and their extraction media can be organic solvents or aqueous solutions. Phytochemical analysis revealed rich nonpolar phytochemical contents in organic extracts and rich polar phytochemical contents in aqueous extracts. However, it is clear that aqueous extracts exhibit good sustainable properties. Each plant extract consists of several major phytochemicals, and these phytochemicals contain a variety of polar functional groups with electron-rich sites, such as hydroxyl (–OH), amino (–NH_2_), amide (–CONH_2_), acid chloride (–COCl), carboxylic acid (–COOH), and ester (–COOC_2_H_5_), which facilitate phytochemical adsorption over the metal surface. Appendix A summarizes the techniques, methods, instruments used, and the results obtained in previous research regarding plant extracts as corrosion inhibitors for carbon steel in 1.0 M HCl media [220,221,222,223,224].

### 4.5. Polymers and Polymeric-Nanoparticles

In recent years, the application of polymers, copolymers, grafted polymers, and polymer composites as green corrosion inhibitors has attracted much attention. Compared with small molecule inhibitors, polymers possess better film-forming capabilities due to their larger size and greater number of functional anchoring groups (nonionic, cationic, anionic, or ampholytic). That is, the anchoring groups of polymers can be easily adsorbed on the metal surface, and polymer inhibitors will more easily achieve a large coverage area than small molecule inhibitors. Polymers’ molecular size, weight, composition, and nature of the anchoring groups (the electronic structure, steric factor, aromaticity, electron density at donor site, and the presence of functional groups such as –CHO, –N=N, R–OH, etc.) are the main factors affecting the corrosion inhibition efficiency. The structural groups for the repeating units of several polymer corrosion inhibitors are summarized in Figure 13.

Epoxy resin is defined as low-molecular-weight pre-polymers with two or more epoxy groups, which is the most common polymeric coating, but there are also examples of studies that investigate the possible use of epoxy resin as corrosion inhibitor for carbon steel in 1.0 M HCl media. In a recent study, Damej et al. synthesized a new organic compound type epoxy resin, namely N, N, 1-tri(oxiran-2-ylmethoxy)-5-((oxiran-2-ylmethoxy)thio)-1H-1,2,4-triazol-3-amine used as corrosion inhibitor for carbon steel C38 in 1.0 M HCl [225,226]. Unexpectedly, through stationary and transiency measurements, the inhibition efficiency of this new epoxy resin reached a value of 92% at only a concentration of 1 mM, and it was also observed that C38 steel surface morphology in the presence of inhibitor remained clean after immersion in 1.0 M HCl, which shows that this epoxy resin protected the C38 steel well against corrosion.

Polyaniline and its derivatives are effective corrosion inhibitors of mild steel in acidic media, which have been well known for years by industry experts. In early studies [227], poly(aniline-formaldehyde) was synthesized from aniline by purification under reduced pressure by Quraishi et al., and then the corrosion inhibition efficiency of poly(aniline-formaldehyde) on mild steel in 1.0 M HCl was evaluated using WL measurements and electrochemical techniques. The results showed that the corrosion inhibition efficiency was over 90% with the addition of only 1 ppm polymer. In another early study, hyperbranched poly(cyanurateamine) was synthesized from diethylene triamine and triallyl cyanurate by Thirumoolan et al. [228]. In their study, the chemical structure and degree of branching of hyperbranched polymer were characterized by FTIR spectroscopy, ^1^H and ^13^C NMR spectroscopy, the number and weight average of the polymer were measured Gel permeation chromatography (GPC). After that, the corrosion inhibition performance of the hyperbranched polymer on mild steel in 0.5~3 M HCl at 25~70 °C was evaluated. Among them, the PDP data showed that the polymer acted as a mixed-type inhibitor with predominant cathodic effectiveness and achieved the maximum corrosion inhibition efficiency (98%) in 1.0 M HCl at 2 mL/L and 25 °C. For the polymers in category polyacryl amide, a few years ago, a similar experimental study was also performed by El-Din et al. [229]. In their experiments, the inhibition performance of mild steel corrosion in 1.0 M solution in the presence of poly(acrylamide-vinyl acetate) (SG) and sulfidated poly(acrylamide-vinyl acetate) (AG) was studied by various measurements. These results indicated that an inhibition efficiency of over 90% of the two copolymers could be achieved at 400 ppm. At the same time, according to the SEM images of the mild steel surface in the absence and presence of SG, the latter could be clearly observed to have improved surface quality, which confirmed the excellent corrosion inhibition effects of the copolymer inhibitors. For the polymers in category polyvinylpyridines, Larabi et al. [230] investigated the synergistic effect between poly(4-vinylpyridine) (P4VP) and potassium iodide (KI) on the corrosion inhibition of mild steel in 1.0 M HCl solutions. Authors found that there are significant differences in the inhibition efficiency of each concentration (1–100 mg/L) of P4VP in presence and absence of 0.1% KI at 25 °C, among them, the values of inhibition efficiency of 1 mg/L P4VP + 0.1% KI and 100 mg/L P4VP were very close (approximately 91%) in the studied metal/electrolysis system, which could be attributed to the fact that the chemisorption of P4VP is stabilized by the addition of 0.1% KI. Moreover, for the polymers in category polyvinyl alcohols, Rahiman et al. [231] evaluated the suitability of poly(vinyl alcohol-cysteine) synthesized from polyvinyl alcohol (PVA, 14,000 g/mol) and ι-cysteine (121.16 g/mol) in 0.5 M oxalic acid by ammonium persulfate treatment as corrosion inhibitor for mild steel in 1.0 M HCl. From the scanning electron micrograph (Figure 14) and the EDX analysis, the presence of small ratio of secondary phase-polymerized cysteine which was randomly distributed and adhered on PVA matrix, and the formation of the composite material and hydrogen bonding is attributed to the force that holds the polycysteine to PVA matrix. The results suggested that poly(vinyl alcohol-cysteine) adsorption obeyed the El-Awady isotherm, and a maximum inhibition efficiency of 94% was obtained in the presence of 0.6 wt% polymer. SEM results demonstrated that the mild steel surface was well protected by the formation of an adsorbed film of poly(vinyl alcohol-cysteine).

In another related study, the corrosion inhibition of mild steel in 1.0 M HCl solution in the presence of two different average molecular weights of photo-cross-linkable polymers (designated as Cl-5c and Cl-10c, corresponding to 11,600 and 12,300 g·mol^−1^, respectively) was investigated by Baskar et al. [232] using electrochemical techniques. PDP and EIS measurements clearly showed that Cl-5c exhibits better inhibitive properties at 15 ppm and 298 K for 2 h (the IE values of Cl-5c and Cl-10c were 99.1% and 97.1%, respectively). This conclusion is not consistent with most of the published research (that is, the inhibition efficiency increases with increasing molecular weight) [233,234], while the explanation given by the authors was the higher degree of photo-cross-linking of the resultant Cl-5 polymer. Moreover, the two photo-cross-linked polymers at low concentrations exhibited good corrosion inhibition efficiency (>90% IE at 5 ppm). In contrast, in the same metal/electrolysis system, higher concentrations (100 ppm) of poly(vinyl alcohol-O-methoxy aniline) and 2-pyridal disulfide were needed to obtain inhibition efficiencies of 95.8% and 65.0%, respectively [74,235]. Carboxymethyl cellulose (CMC) is another research focus which characterized by a high molecular weight polymer with water-soluble. In a study by Bayol et al. [236], the effect of sodium carboxymethyl cellulose (Na-CMC) on the corrosion behavior of mild steel in 1.0 M HCl solution was investigated by using the WL, EIS and LPR methods. The results showed that this water-soluble polymer is a mixed-type inhibitor, its adsorption on the CS/HCl interface followed the Langmuir adsorption isotherm, and the maximum inhibition efficiency of 78% was achieved at 0.04%. 

Polymer nanoparticles and inorganic (metal/metal oxide) nanoparticles can also be used as additives for inhibiting corrosion because of their great surface-to-volume proportion. The mechanism of the inhibition is explained as follows: nanoparticles exposed to a corrosive environment can form self-adhering films (as a protective film) on the surface of the metal substrate, which can reduce the corrosion rate by blocking the active sites over the metal surface [237]. It is worth mentioning that the surface stability of nanoparticles is an important factor in maintaining their corrosion inhibition performance, while the protective film of inorganic nanoparticles is unstable, thus, inorganic nanomaterials are usually functionalized by introducing different functional groups. Polymers are an ideal choice for the functionalization of inorganic nanoparticles because these compounds, as stable corrosion inhibitors, exhibit good corrosion inhibition performance for metallic materials in acidic media. In general, inorganic nanomaterials can be chemically bonded with polymer matrices to form metal complexes, and they can also be dispersed in polymer matrices. 

For polymer nanoparticles, in an early study by Atta et al. [238], amphiphilic chitosan (CS) nanogels were prepared by using a surfactant-free method, that is, that unsaturated fatty acids (oleic acids (OA) and linolenic acids (LA)) were introduced into CS as hydrophobic chains, and then polyethylene glycol (MPEG550-CHO aldehyde) was introduced into CSOA and CSLA as hydrophilic chains (the two derivatives were designated as CSOA-MPEG and CSLA-MPEG). The influence of the two nanogels on the corrosion of steel in 1.0 M HCl was tested by electrochemical and contact angle measurements. The results showed that both CSLA-MPEG and CSOA-MPEG showed excellent corrosion inhibition performance at low concentrations (>90% IE at 50 ppm), which was attributed to the crosslinking and interaction between the nanogels and steel surfaces. Among both nanogels, a well-adsorbed film was formed on the steel surface at lower concentrations of CSLA-MPEG, which is interpreted as a consequence of the higher surface activity of CSLA-MPEG. From the contact angle data, the adsorbed film containing the nanogel inhibitors decreased the active surface area of the steel exposed to an acidic media. In another similar study, polydopamine nanoparticles (PDAs) with different water stabilities, sizes, and chemistries (PDA-1, PDA-2, and PDA-3) were synthesized by Habibiyan et al. [239] using three different methods. After that, the stabilities, film-forming capabilities, and corrosion inhibition performances of the three polydopamine nanoparticles (in mild steel/1.0 M HCl system) were compared. Among them, PDA-2, with the highest number of surface charges, showed the best stability in a neutral pH aqueous solution by zeta potential and stability tests. Based on SEM images taken in this study (Figure 15), PDA-2 nanoparticles have a lower particle size and better dispersibility, which cause them to form a more uniform film on the steel surfaces in acidic solution. As expected, PDA-2 with open rings and full carboxylic acid groups showed the best corrosion inhibition performance (approximately 99% IE at 5 mg/L). From the amount of inhibitor usage and the inhibition efficiency perspective, these polydopamine nanoparticles were much better than chitosan and alginate (91% IE at 1500 mg/L and 83% IE at 0.1 mg/L, respectively) [240,241]. Additionally, an interesting study reported the possible exploitation of NanoCars for corrosion protection purposes [242]. NanoCars are molecular machines capable of transferring efficiently with some control and directionality at the nanoscale level onto the metal surface by using an advanced scanning tunnelling microscope. In that study, the authors assessed the adsorption ability, adsorption centers, geometry, and adsorption energetics of NanoCars onto the Fe(110) interface by using using several theoretical methods (DFT, MC, and MD) based upon molecular level details. The results demonstrated that NanoCars molecules flat-lay onto the iron surface, and the enormous adsorption energies are supportive of a form interaction of the responsible adsorption sites (O and N atoms) of NanoCars with the iron surface, which form a barrier film that slows the diffusion of the corrosive species toward the metal surface.

For inorganic and organic-based nanoparticles, Azzam et al. [243] synthesized poly 12-(3-amino phenoxy) dodecane-1-thiol surfactant (C12P), silver nanoparticles (AgNPs), and the nanostructure of the synthesized polymeric thiol surfactant self-assembled on AgNPs (C12P + AgNPs). Based on TEM investigation, the presence of the alkyl chain of C12P increased the stability of AgNPs and decreased the aggregation of AgNPs due to the interaction with these surfactant molecules. Based on WL and PDP measurements, both C12P and C12P + AgNPs at 375 ppm are effective corrosion inhibitors of carbon steel in 1.0 M HCl solution, while C12P + AgNPs showed a slightly higher corrosion inhibition efficiency than C12P due to a stronger adhesion of their protective film on the carbon steel surface. In another comparative study by El-Lateef et al. [244], TiO_2_ nanofiber (TiO_2_ NFs) Schiff base phenylalanine (SBP) composite (TiO_2_ NFs/SBP) were designed. Figure 16 shows the SEM images of the pristine [TiO_2_ NFs] and [TiO_2_ NFs/SBP] composites, from which [TiO_2_ NFs/SBP] contained fibers and platelet-like structures with variable dimensions. After the structural characterization of the composites, the effects of C-steel on corrosion inhibition in 1.0 M HCl in the absence and presence of different [TiO_2_ NFs/SBP] additions were compared using various electrochemical methods. The findings showed that the corrosion inhibition efficiency rises with increasing [TiO_2_ NFs/SBP] addition, and the highest inhibition efficiency of 97.9% was obtained at 300 mg/L. Moreover, the adsorption of the composite was determined to follow a Langmuir isotherm, and the value of Gibbs free energy (ΔG_ads_) supported the chemisorption and physisorption mechanisms for the composite.

The abovementioned studies take advantage of the innate features of polymer molecules, such as highly versatile derivatization, a large number of binding sites, and better film-forming properties to stabilize and/or carry out their corrosion inhibition actions. In the early years (2009–2014), polymers as corrosion inhibitors were reviewed by Umoren, Arthur, Sabirneeza, and Tiu et al. [245,246,247,248]. Recently, Shahini et al. [249] also reviewed the progress of biopolymer-type inhibitors. It can be seen that polymers as corrosion inhibitors have received much attention from scientific researchers. However, compared to small molecule inhibitors, the application of polymeric inhibitors is uncommon in the actual production environment. This is clearly because it is difficult to achieve efficient and controlled polymerization of desired polymers with simple polymerization techniques. On the other hand, there may be a disadvantage of poor dispersion under some conditions. Appendix A summarize the techniques, methods, instruments used, and the results obtained in previous research regarding polymers and nanoparticles as corrosion inhibitors for carbon steel in 1.0 M HCl media, respectively.

### 4.6. Statistical Analyses

For further analysis, we compared and analyzed the corrosion inhibition performance for six kinds of inhibitors using the histogram of max inhibition efficiency values from Appendix A (providing the mode of these IE values), which contain their normal distribution curve (providing the mean of these IE values), as shown in Figure 17a. At the same time, the box-whisker plots were also plotted against these IE values, which marked the distributions of the median, the lower quartile (splits 25% of lowest data), the upper quartile (splits 75% of highest data), and the minimum and maximum values (diamond dots denote the mean) (Figure 17b).

Based on the two statistics charts, the maximum inhibition efficiency distributions for six groups of inhibitors can be easily and clearly comparable, in which surfactants, polymers, and nanoparticles as corrosion inhibitors show a higher inhibition performance for carbon steel corrosion in 1.0 M HCl media. The distribution of these values is concentrated at 94~98% for surfactant inhibitors, 94~100% for polymer inhibitors, and 96~98% for nanoparticle inhibitors, and the overall inhibition efficiency of ionic liquid inhibitors is slightly lower than that of the other five groups of inhibitors in this analysis. On the other hand, the lower quartile of IE for ionic liquid inhibitors is under 90%, while all other five groups of inhibitors are larger than 90%. For surfactant inhibitors, polymers inhibitors, and nanoparticle inhibitors, they all exceeded 92%, that is, most of them (75%) focus in a high IE range of 92~100%. Therefore, these three inhibitor types are ideally suited for inhibiting carbon steel corrosion in 1.0 HCl solution.

The inhibitor concentrations used are an important aspect in corrosion inhibition performance. Generally, increasing inhibitor concentrations can increase the surface coverage of metals, which is likely to lead to an increase in protection efficiency. According to the collected literature information, the inhibition efficiency of most of the inhibitors at optimum concentrations of corresponding study can achieve or approach the maximum, that is, that a further addition of inhibitor does not cause any appreciable increase in the inhibition efficiency. However, a few studies have demonstrated that a further increase in the inhibitor concentration beyond the optimum concentration causes a decrease in the inhibition efficiency [250,251,252]. This condition is usually interpreted to mean that an excess in the inhibitor concentration results in the aggregation and/or precipitation of inhibitor molecules due to the enhancement of the intermolecular force, which can decrease the adsorption tendency over metallic surfaces and ultimately the inhibition efficiency [101].

Among the inhibitors listed, most of the optimum concentration values are given in moles/millimoles/micromoles per liter. For ease of comparison, their corresponding concentration values and their corresponding maximum inhibition efficiency were screened from Appendix A, and then a scatter plot with these optimum concentration and maximum inhibition efficiency was plotted, as shown in Figure 18. In the scatter plot, the *X*-axis and *Y*-axis indicate the optimum concentration and maximum inhibition efficiency, respectively, while each point represents one study result. Among them, the yellow dots represent the data for polymer inhibitors, a large fraction of which are concentrated in the upper left corner region, and this is an ideal region with a low concentration/high inhibition efficiency (as shown by the yellow shaded region in Figure 18). The blue dots denote the data for surfactant inhibitors, which are mainly concentrated in the upper-middle region with high inhibition efficiency (as shown by the blue shaded region in Figure 18), but their optimum concentrations used are slightly higher than those of polymer inhibitors. It can be inferred that a higher concentration of surfactant inhibitors is needed to reach the same coverage areas as polymer inhibitors due to the molecular size differences between the two inhibitors. In this portion of the analysis, the above two inhibitor types are ideally suited for inhibiting carbon steel corrosion in 1.0 HCl solution.

The red and green dots represent the data for nanoparticle and plant extract inhibitors, respectively, most of which are grouped close to the center in the scatter plot, especially plant extract inhibitors (as shown by the red and green shaded regions in Figure 18). The local enrichment of dots representing the plant extract inhibitors indicates that a variety of plant extracts as well as the plant extracts obtained using the various extraction methods have similar inhibition performance (especially the optimum inhibition concentration), which seems to be an inherent property of these inhibitor types. This is most likely because the main active ingredients of these plant extracts for inhibiting carbon steel corrosion in 1.0 HCl solution share similar chemo-physical properties. Additionally, it is evident from the red and green shaded regions that there is a lower overall inhibition performance for the two inhibitor types than for the polymer and surfactant inhibitor types. 

The gray and pink shaded regions indicate the areas of the majority of dots representing the drug inhibitors (gray dots) and ionic liquid inhibitors (pink dots), respectively (as shown in Figure 18). It is apparent that the two groups of dots have larger distribution areas compared to the other four groups (larger regions correspond to larger within-group differences). According to the locations of the shaded areas, it can be judged that the overall inhibition performances of the two inhibitor types are also lower than those of the polymer and surfactant inhibitor types.

In addition, temperature is also an important factor to consider when selecting corrosion inhibitors for carbon steel/1.0 HCl solution system. Usually, in the case of physisorption, as the temperature increases, the inhibition performance decreases, which is attributed to the increase in the kinetic energy of the inhibitor molecules due to the increase in temperature, which in turn enhances the desorption of adsorbed inhibitors from the metallic surface [253]. For chemisorption, at high temperature, electrolyte-catalyzed decomposition, fragmentation, and/or rearrangement of the inhibitor molecule can decrease or increase (through synergism) the inhibition efficiency of the inhibitors [36,253]. According to the information extracted from relevant literature (Appendix A), most of these inhibitors give the corresponding temperature at which the maximum inhibition efficiency is achieved. The following pie charts (Figure 19) summarize the number and the percentage of inhibitors at the indicated range of temperatures for each group. The temperature ranges of <30 °C and 30~49 °C account for a major percentage of the total number of inhibitors in each group, which are the most common environmental conditions of inhibitors used for carbon steel/1.0 HCl solution system. A small proportion of inhibitors have an optimum temperature range of >50 °C (except for the drug inhibitors), and surfactant inhibitors make up slightly more than half in this group and contain one of the highest optimum temperatures (80 °C). Therefore, surfactants may be better suited to serve as potential inhibitors used at high temperatures. In contrast, drug inhibitors are most probably not suited to such potential applications.

## 5. Conclusions

In this review, the corrosion and inhibition mechanisms of carbon steel/HCl solution systems, 12 different methods for the evaluation of corrosion inhibition efficiency, 7 different adsorption isotherm models, multiple different adsorption thermodynamic parameters, QM calculations (including several commonly used global and local reactivity descriptors), MD/MC simulations, and the main characterization techniques used were comprehensively introduced and summarized, which cover the majority of crucial elements for the evaluation of corrosion inhibition performance in recent years. This section will be helpful in determining the study protocols of corrosion inhibitors quickly for beginners. 

In the classification and statistical analysis section, organic compounds or (nano)materials as corrosion inhibitors were divided into six taxonomic groups: drug molecules, ionic liquids, surfactants, plant extracts, polymers, and polymeric nanoparticles. The statistical analysis was only carried out using data from carbon steel/1.0 M HCl solution system. By comparing six types of inhibitors, polymers with larger molecular sizes can be used as better corrosion inhibitors for carbon steel/1.0 HCl solution system, which can cover the larger metallic surface area at low concentrations compared to compounds with smaller molecular sizes. On the other hand, a greater number of functional anchoring groups is an important factor affecting the electron density over the active sites and the geometry of the inhibitor molecules over the metallic surface (by adjusting the nonbonding and π-electron conjugation and steric hindrance), which may help improve the corrosion inhibition efficiency of polymers. Furthermore, the overall inhibition performance of surfactant types is also relatively ideal, which further indicates that the substituents effect (the hydrocarbon chains act as hydrophobic moieties in their structures, as mentioned above in Section 2.2) may also have played a role. In this regard, the current research items specific to surfactant inhibitors include surface tension, conductivity, zeta potential, emulsification, and biodegradability, which are used to test the solubility (hydrophilicity and hydrophobicity) of the inhibitor molecule in specific electrolytes and determine aggregation and colloidal stability. In these examples of this review, for drugs, ionic liquids, plant extracts, and nanoparticles are corrosion inhibitors for carbon steel in 1.0 HCl solution, a few of which also show outstanding inhibition performance at low concentrations. Of course, the low concentration/high inhibition efficiency is not the only measure of the goodness of the inhibitors, and the nontoxic properties (for some drug inhibitors) and the simple and cost-economic synthesis process (for some ionic liquid inhibitors and nanoparticle inhibitors) also need to be considered. Thus, plant extracts, as environmentally friendly corrosion inhibitors, possess great application potential and have been widely investigated. In our statistical analysis, plant extracts’ corrosion inhibition performance (especially the optimum inhibition concentration) did not vary substantially according to plant species and extraction methods. We believe that the main active ingredients of these plant extracts for inhibiting carbon steel corrosion in 1.0 HCl solution share similar chemo-physical properties. Finally, we want to emphasize once again that the data used in this analysis are provided in Appendix A, and obviously, the statistics have limited quality due to the small numbers of studies. However, we believe that this information still has scientific reference value for guiding anti-corrosion options for a “carbon steel/1.0 M HCl” system and can enlighten future research directions.

## Figures and Tables

**Figure 1 materials-15-02023-f001:**
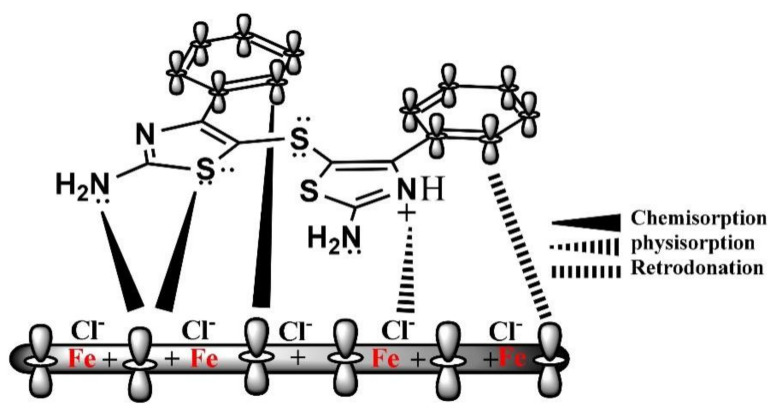
Diagrammatic illustration of the three adsorptions (physisorption, chemisorption, and retro-donation mechanisms) of organic corrosion inhibitors (DHATs). (Reprinted with permission from Ref. [45]. Copyright 2015 Elsevier Publications).

**Figure 2 materials-15-02023-f002:**
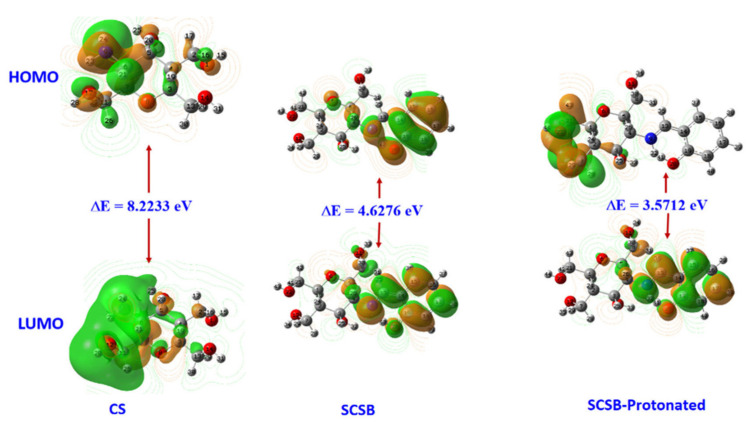
HOMO orbital, LUMO orbital and ΔE of the three inhibitors (chitosan (CS), neutral (SCSB), and pronated salicylayde-chitosan Schiff Base (SCSB-Protonated)). (Reprinted with permission from Ref. [100]. Copyright 2020 Elsevier Publications).

**Figure 3 materials-15-02023-f003:**
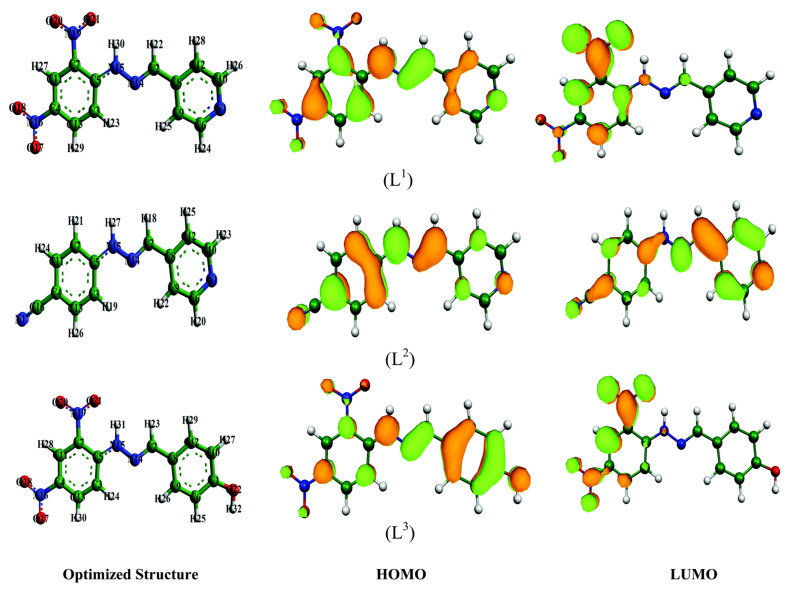
Optimized geometry structures, HOMO and LUMO plots of the three Schiff-base molecules calculated by the B3LYP method, with the SV(P) and SV/J levels of the basis set. (Reprinted with permission from Ref. [111]. Copyright 2016 Royal Society of Chemistry Publications).

**Figure 4 materials-15-02023-f004:**
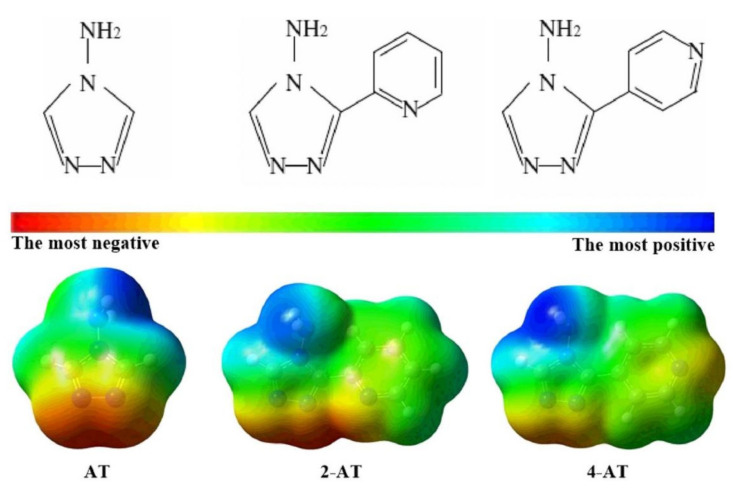
MEP maps of the three inhibitors (2-amino-1,3,4-triazole (AT), 3-(2-pyridyl)-2-amino-1,3,4-triazole (2-AT) and 3-(4-pyridyl)-2-amino-1,3,4-triazole (4-AT)) calculated by B3LYP/6-311 + (2d,p) level. (Reprinted with permission from Ref. [113]. Copyright 2016 Elsevier Publications).

**Figure 5 materials-15-02023-f005:**
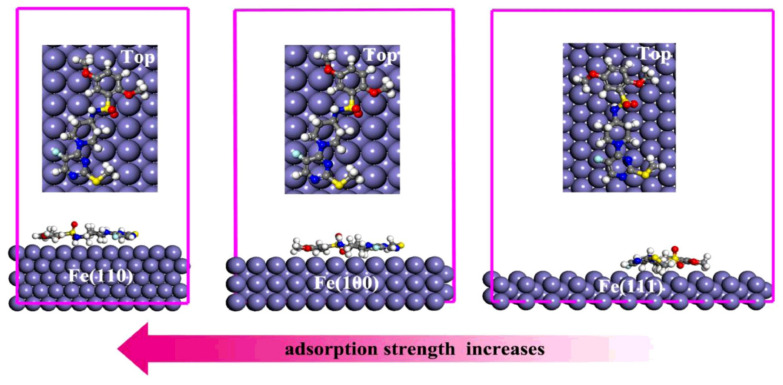
Mode of adsorption of (5-fluoro-2-(methylthio)pyrimidine-4-yl)(piperidine-4-yl)-2,5-dimethoxybenzene sulfonamide (FMPPDBS) on different iron surfaces. (Reprinted with permission from Ref. [127]. Copyright 2016 Elsevier Publications).

**Figure 6 materials-15-02023-f006:**
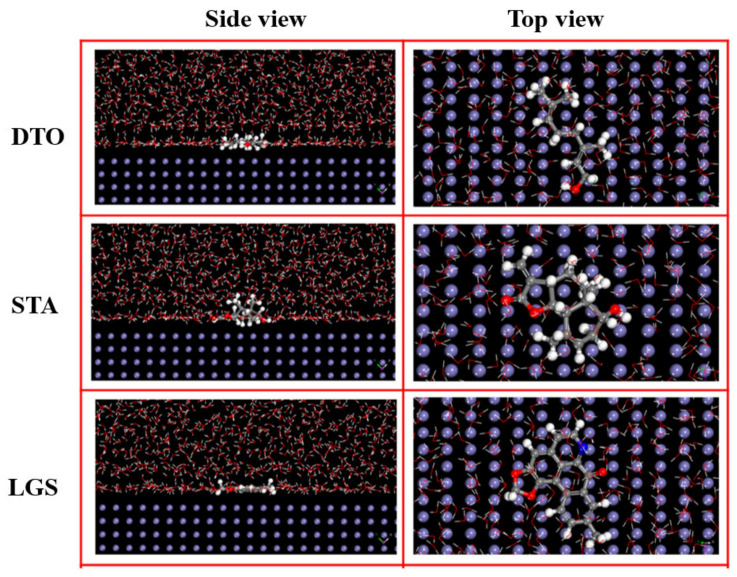
The stable adsorption configurations of DTO, STA, and LGS molecules on the Fe (110) surface. (Reprinted with permission from Ref. [130]. Copyright 2020 Elsevier Publications).

**Figure 7 materials-15-02023-f007:**
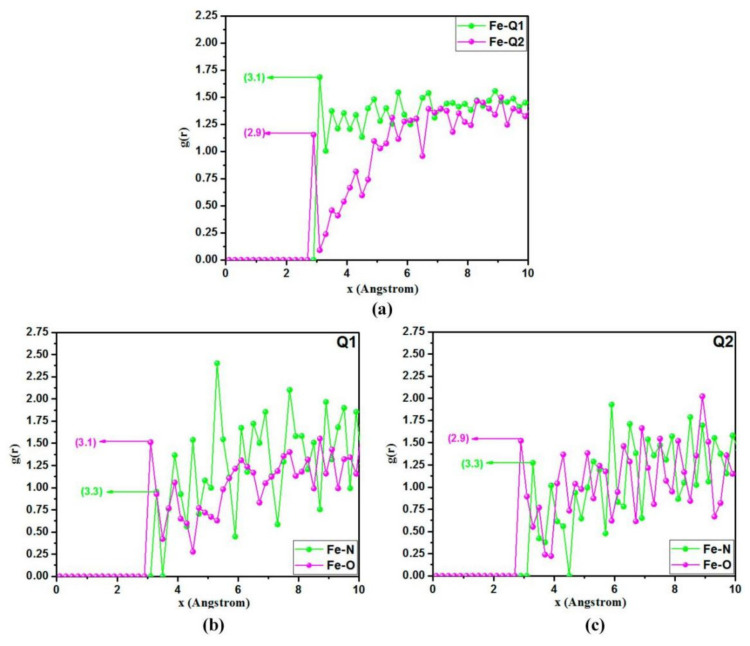
RDF analysis of two quinoxaline derivative inhibitor molecules (Q1 and Q2) on the Fe (110) surface in simulated solution, (**a**) Fe-heteroatoms of Q1, (**b**) Fe-heteroatoms of Q2, and (**c**) Fe-inhibitors. (Reprinted with permission from Ref. [134]. Copyright 2020 MDPI Publications).

**Figure 8 materials-15-02023-f008:**
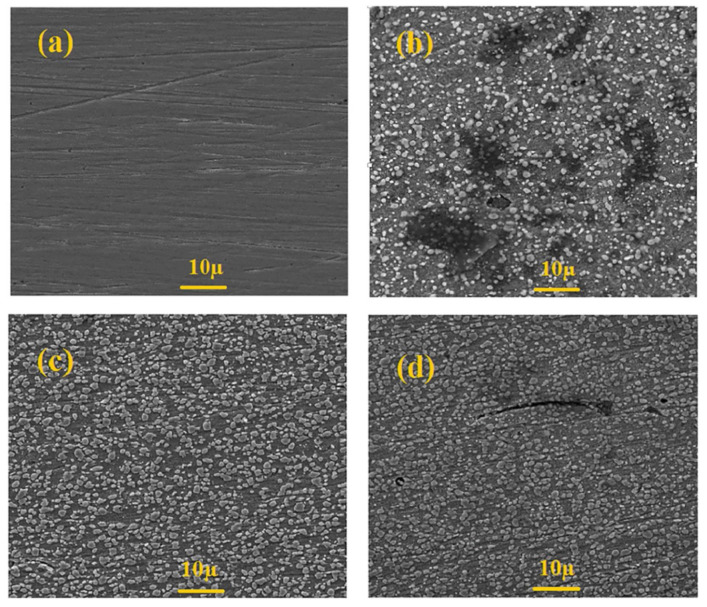
SEM image of mild steel surface, (**a**) freshly polished mild steel surface, (**b**) mild steel surface after 4 h of soaking in 1.0 M HCl, (**c**,**d**) mild steel surface after 4 h of soaking in 1.0 M HCl in the presence of 5 mM L-cysteine and D-penicillamine, respectively. (Reprinted with permission from Ref. [139]. Copyright 2020 Elsevier Publications).

**Figure 9 materials-15-02023-f009:**
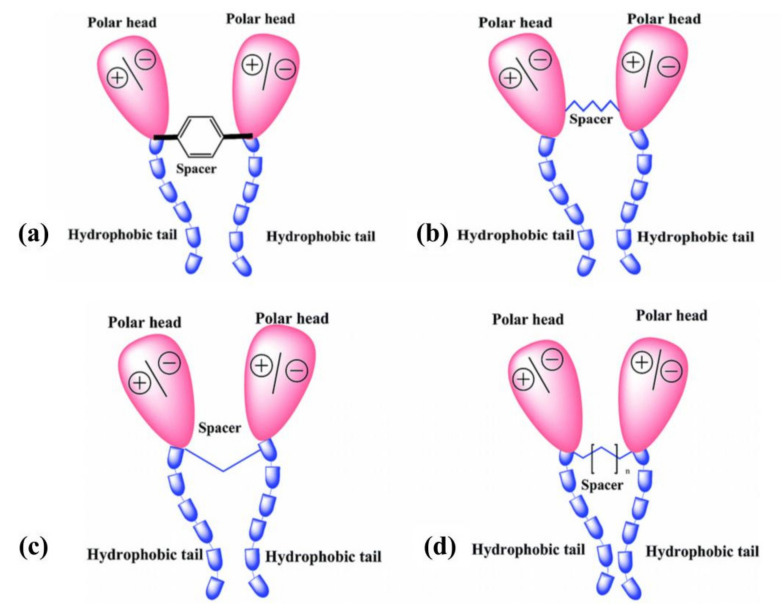
Simplified schematic representation of the physical structure of gemini surfactant: (**a**,**b**) are gemini surfactants with rigid and flexible spacers, (**c**,**d**) are gemini surfactants with short chain and long chain spacers, (**e**,**f**) are gemini surfactants with polar and nonpolar spacers, and (**g**,**h**) are gemini surfactants with two identical and nonidentical hydrophobic chains. (Reprinted with permission from Ref. [203]. Copyright 2015 Royal Society of Chemistry Publications).

**Figure 10 materials-15-02023-f010:**
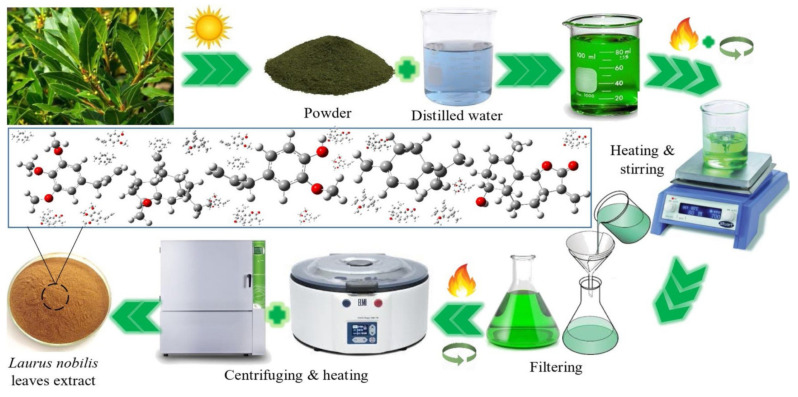
Schematic route for preparing Laurus nobilis leaf extract powder that was used for mild steel corrosion prevention. (Reprinted with permission from Ref. [213]. Copyright 2020 Elsevier Publications).

**Figure 11 materials-15-02023-f011:**
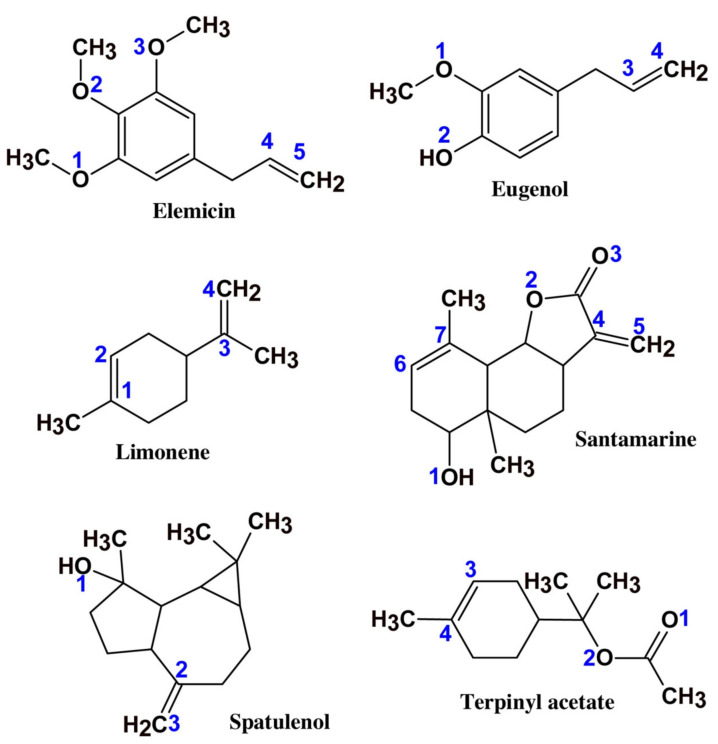
Geometry of neutrally charged elemicin, eugenol, limonene, santamarine, spatulenol, and terpinyl acetate compounds found in Laurus nobilis leaf extract, the oxygen and carbon atoms chosen for protonation are indicated in blue. (Reprinted with permission from Ref. [213]. Copyright 2020 Elsevier Publications).

**Figure 12 materials-15-02023-f012:**
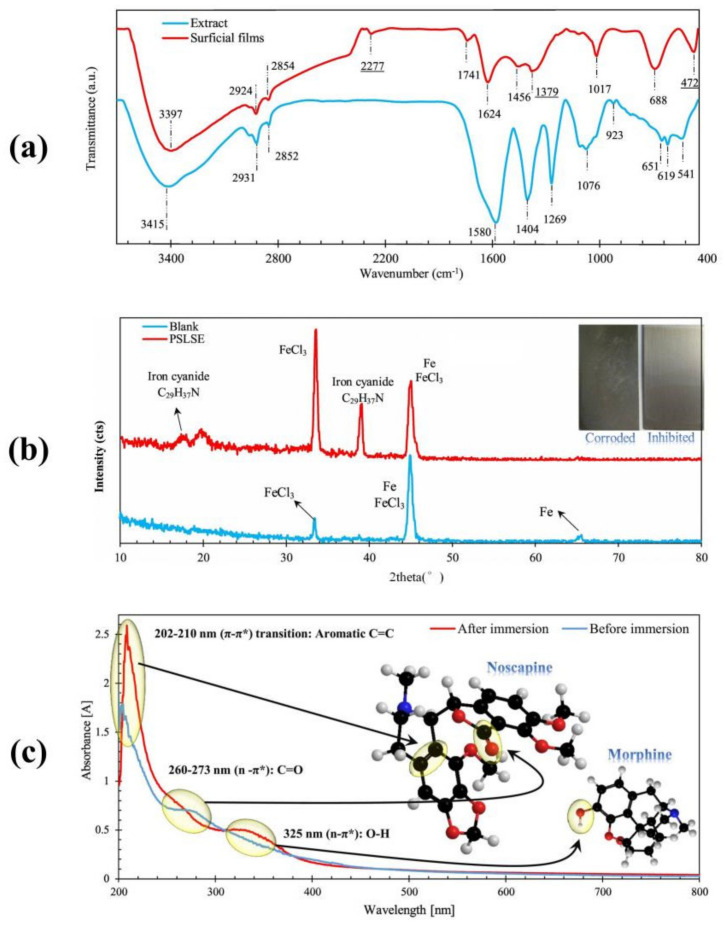
(**a**) FTIR spectra of the extract powder and the surficial film formed on the inhibited steel surface, (**b**) XRD patterns for the steel samples immersed in 1.0 M HCl solution in the presence and absence of PSLSE, (**c**) UV–Vis spectra before and after immersing the steel samples into 1.0 M HCl solution in the presence of PSLSE for 2.5 h. (Reprinted with permission from Ref. [215]. Copyright 2020 Elsevier Publications).

**Figure 13 materials-15-02023-f013:**
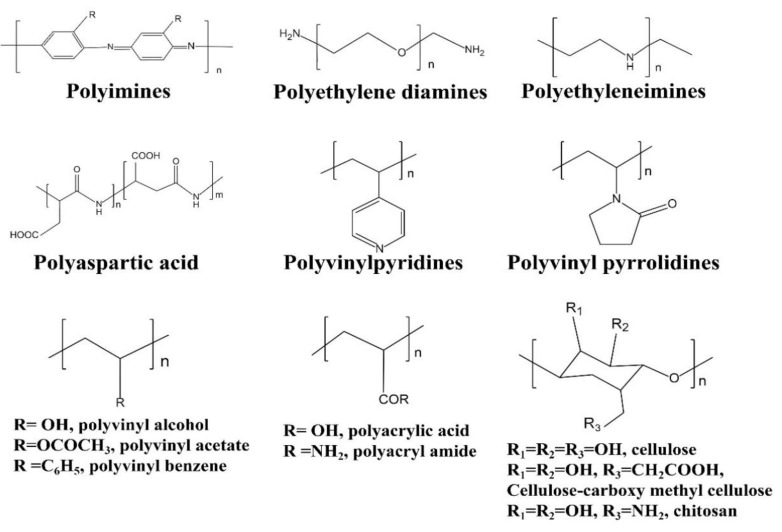
Structural groups of polymer inhibitors.

**Figure 14 materials-15-02023-f014:**
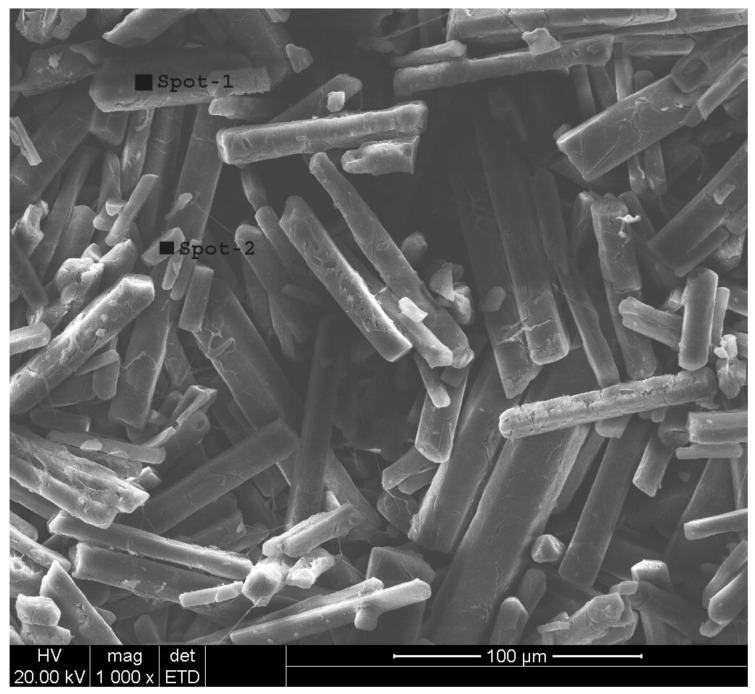
Scanning electron micrograph of poly(vinyl alcohol-cysteine). (Reprinted with permission from Ref. [231]. Copyright 2017 Arabian Journal of Chemistry Publications).

**Figure 15 materials-15-02023-f015:**
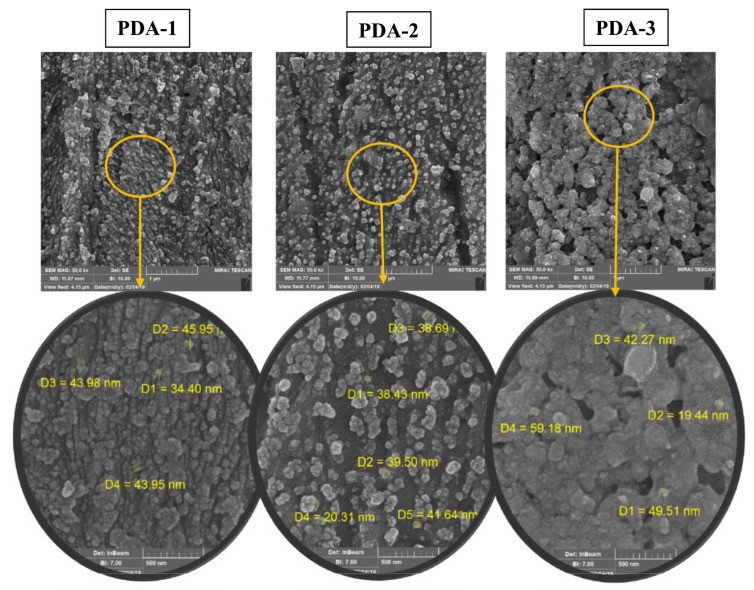
SEM images of mild steel surfaces after 24 h immersion in 1.0 M HCl solution in the presence of PDA-1, PDA-2, and PDA-3. (Reprinted with permission from Ref. [239]. Copyright 2020 Elsevier Publications).

**Figure 16 materials-15-02023-f016:**
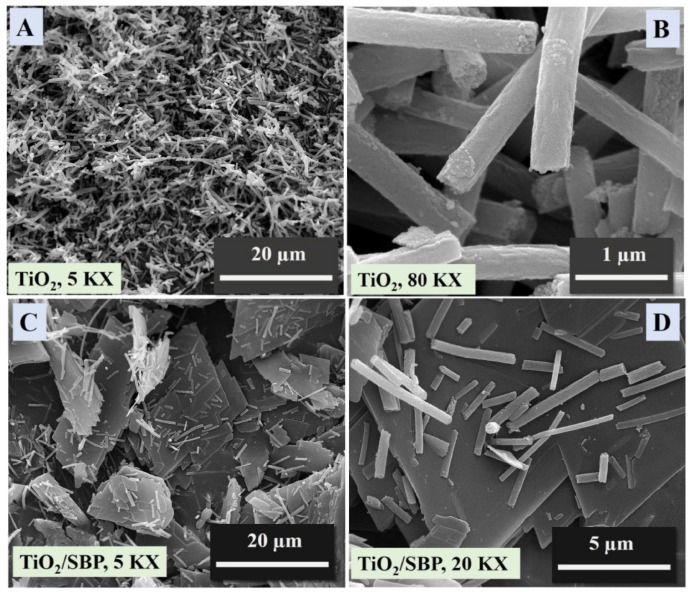
SEM images of [TiO_2_ NFs] (**A**,**B**) and [TiO_2_ NFs/SBP] (**C**,**D**). (Reprinted with permission from Ref. [243]. Copyright 2013 Egyptian Journal of Petroleum Publications).

**Figure 17 materials-15-02023-f017:**
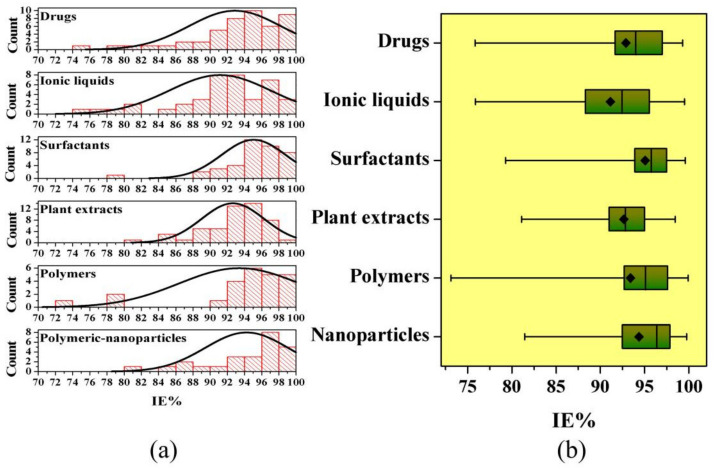
(**a**) Histogram of maximum inhibition efficiency values for six kinds of inhibitors, (**b**) box-whisker plots of maximum inhibition efficiency values for six kinds of inhibitors, the source of the data is in Appendix A.

**Figure 18 materials-15-02023-f018:**
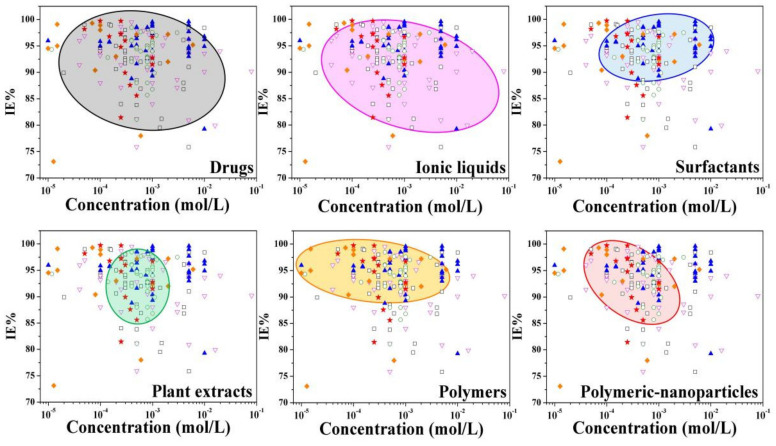
Scatter plot of optimum concentration and the corresponding max inhibition efficiency for six kinds of inhibitors, and their respective the main region of the distribution (the source of the data is in Appendix A where the optimum concentration values are given in moles/millimoles/micromoles per liter).

**Figure 19 materials-15-02023-f019:**
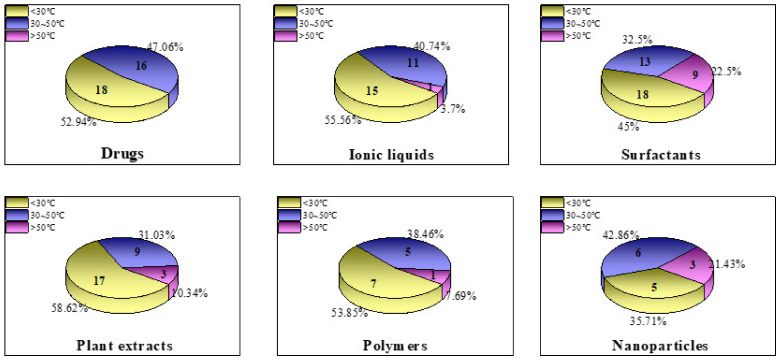
The number and the percentage of each type of inhibitor at the indicated range of temperatures that achieved the maximum inhibition efficiency. The source of the data is in Appendix A.

**Table 1 materials-15-02023-t001:** Corrosion rates for mild steel samples at different concentrations of HCl solutions at 25 °C.

*c*_HCl_ (mol·dm^−3^)	0.25	0.5	1.0	1.5	2.0	2.5
*ρ*_HE_ × 10^2^ mL·cm^−2^·min^−1^	2.083	3.308	5.157	5.658	7.645	9.106
*ρ*_ML_ × 10^5^ g·cm^−2^·min^−1^	5.841	8.143	12.214	13.060	17.756	20.986

**Table 2 materials-15-02023-t002:** Methods used for evaluating corrosion inhibition efficiency.

Evaluation Methods	Abbreviations
Weight loss measurements	WL
Potentiodynamic polarization	PDP
Electrochemical impedance spectroscopy	EIS
Linear polarization resistance	LPR
Potentiodynamic anodic polarisation	PDAP
Cyclic voltammetry	CV
Electrochemical noise	EN
Electrochemical frequency modulation	EFM
Scanning vibrating electrode technology	SVET
Potential of zero charge	PZC
Hydrogen evolution measurements	HE
Thermometric methods	-

**Table 3 materials-15-02023-t003:** The six adsorption isotherm models.

Number	Adsorption Isotherm Model	Equation	Reference
No.1	Langmuir isotherm relation	$\frac{C}{\theta }=\frac{1}{K_{}}+C$	Ref. [80]
No.2	Temkin isotherm relation	$\log\frac{\theta }{C}=\mathrm{logK}+g\theta$	Ref. [81]
No.3	Freundlich isotherm relation	$\log\theta =\log K+\frac{1}{n}\log C$	Ref. [82]
No.4	Frumkin isotherm relation	$\log\left[\frac{\theta }{(1-\theta )C}\right]=\mathrm{logK}+g\theta$	Ref. [81]
No.5	El-Awady isotherm relation	$\log(\frac{\theta }{1-\theta })=\log K+y\log C$	Ref. [83]
No.6	Flory–Huggins isotherm relation	$\log\frac{\theta }{C}=\log K+n_{FH}\log(1-\theta )$	Ref. [84]

## Data Availability

No new data were created or analyzed in this study. Data sharing is not applicable to this article.

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
