# Peer review of "Organic Compounds as Corrosion Inhibitors for Carbon Steel in HCl Solution: A Comprehensive Review"

_materials, 2022, doi:10.3390/ma15062023_

Round 1

Reviewer 1 Report

The manuscript concern  an organic corrosion inhibitors for CS / HCl system review.

The manuscript is an interesting attempt to organize and compare organic corrosion inhibitors. The manuscript was well prepared, the proposed issues are proper.

This manuscript will be a valuable source of knowledge, especially when introducing the reader to the subject.

The work is extensive, with a large amount of reference which nevertheless takes up a large part of the manuscript.

The manuscript was well prepared, it contains almost no errors, I have no critical remarks for the text.

Only Figure 9 needs to be improved: “a” and “b” parts are invisible, "h" covers the description. Please correct.

In addition, I did not have access to Supplementary material and could not evaluate this item.

Reviewer 2 Report

The present paper represents a review about corrosion inhibitors of carbon steel with a comprehensive background about the subject. I do not see any good contribution of this review to the subject. There are plenty of similar review on different kind of corrosion inhibitors in different mediums. I do not see a need for a new one in Carbon steel-HCl interface. A new review should be about a subject that lack of sufficient review. Therefore, I do recommend the rejection.

Reviewer 3 Report

1. Title
   Consider the following suggestion: " A comprehensive evaluation of organic molecules and (nano)materials as 
corrosion inhibitors for carbon steel in 1 M HCl aqueous solution "

3. Simplify the following paragraph " By collating information from recent literature, most studies on the corrosion inhibition performance of organic inhibitors were conducted within “carbon steel/1.0 M HCl” solution
system, and their research methods (experimental and theoretical methods) used were also similar to each other, with some slight variations."

4. Please, provide the first use of an abbreviation immediately in the text.

i.e  QC calculations, MD/MC  .. . . .

5. When disscusing the theoretical calculations, keep in mind the:
 Molecular Mechanic - MM (quantum mechanic: MC or MD) from Quantum  Mechanic QM (DFT...)

6. Why "used for organic inhibitor" these are general techniques not only for "organic" inhibitors

This review article will first briefly describe the different experimental techniques
and theoretical analysis methods widely used for organic inhibitor research in the last
few years. 

7. Re-write  
"Finally, the
critical information and contrastable experimental data from different studies on each
type of corrosion inhibition in carbon steel/1.0 M HCl solution system will be enumerated
and listed, and use them to contrasts the differences in corrosion inhibition performance
between these types of organic compounds, especially in their maximum inhibition efficiency, optimum concentration and optimum temperature. "
same for "This work can be helpful in
determining research methods quickly for beginners. In practice work, it is able to provide a clearer choice of anti-corrosion options for this system"

8. Omit Bold "Mulliken charge distribution can also be used to determine the active sites of the reaction between an inhibitor molecule and a metal surface. In general, the more negative
the atomic charge of the adsorption sites is, the easier it would be to donate electrons
to nearby unoccupied orbitals. The Mulliken charge distribution is often combined
with the HOMO energy density distribution to predict the active site of the inhibitor
molecule, i.e., the more negative the atomic charge is and the higher the HOMO energy density is, the more likely the adsorption site is."

9. Equation 65, there are different expressions for different force fields, thus some of the common force field used in corrosion can be compared so the readers can see 
how they are mathematically expressed ( i.e Universal vs. COMPASSII)

10. Simplify  "After MD simulations, the calculations of the radial distribution function (RDF) g(r)
can be further used to extract information about the bond length and the type of interaction of inhibitor molecules on the metal surface from a numerical simulation, which is a
useful tool to evaluate the MD results."

11. Carefully analyze the content in the Table 3 

    there are given instrumental techniques not wat is obtained i.e High resolution mass spectrum ??? /// High resolution mass spectroscopy or Mass Spectrometry

12. Figure 9 is missing in some parts

13. There are interesting/imortatnt studies (plant extracts, polymers, . . . )  that are not included in this review:
i.e
https://www.sciencedirect.com/science/article/abs/pii/S0022286022000989
https://www.sciencedirect.com/science/article/abs/pii/S2210271X21001171?via%3Dihub

14. Correct "Figure 10. Schematic preparation process of Laurus nobilis leaf extract green ..  ."

15. The LINK to supplementary materials "According to the information extracted from relevant literature (Supplementary Table 4−9),
 most of these inhibitors give the corresponding temperature at which the maximum inhibition" is wrong which didn't allow as to see the data.

Round 2

Reviewer 2 Report

The authors significantly improved the manuscript and made it suitable for the journal scope by modifying the title and new content. However, the main issue I've raised is that I don't favor repeated works. I do see plenty of similar reviews that cover all kinds of corrosion inhibitors in different mediums. A few years later may be suitable for a new one.

Reviewer 3 Report

The version of the paper is of sufficient quality and interest, therefore, we feel like  it  can  be regarded for the possible publication.
